# CausNVS: Autoregressive Multi-view Diffusion for Flexible 3D Novel View Synthesis

## Abstract

Multi-view diffusion models have shown promise in 3D novel view synthesis, but most existing methods adopt a non-autoregressive formulation. This limits their applicability in world modeling, as they only support a fixed number of views and suffer from slow inference due to denoising all frames simultaneously. To address these limitations, we propose CausNVS, a multi-view diffusion model in an autoregressive setting, which supports arbitrary input-output view configurations and generates views sequentially. We train CausNVS with causal masking and per-frame noise, using pairwise-relative camera pose encodings (CaPE) for precise camera control. At inference time, we combine a spatially-aware sliding-window with key-value caching and noise conditioning augmentation to mitigate drift. Our experiments demonstrate that CausNVS supports a broad range of camera trajectories, enables flexible autoregressive novel view synthesis, and achieves consistently strong visual quality across diverse settings. Anonymous project page: https://causnvs.github.io/.

## 1. Introduction

World models (Parker-Holder et al., 2024; Agarwal et al., 2025; Labs, 2024; Russell et al., 2025; Ball et al., 2025) have emerged as a new paradigm that equips agents with internal representations of their environments, enabling them to understand, simulate, and interact with the world. A core capability of such models is spatial intelligence (Davison, 2018), the ability to perceive 3D structure, infer spatial relationships from sparse observations, and generate photorealistic, geometrically consistent views from novel poses. In interactive or exploratory scenarios, this ability must operate sequentially and causally: as the agent moves or receives new pose queries, the system should synthesize views based only on previously accumulated information. This motivates the task of novel view synthesis (NVS), where the goal is to predict unseen views given a sparse set of posed inputs. Recent advances in multi-view diffusion models (Watson et al., 2024; Gao* et al., 2024; Zhou et al., 2025b; Yu et al., 2024a) have significantly improved NVS performance by incorporating large-scale pretrained 2D generative priors. However, these models are inherently non-causal, as they assume all target poses are known in advance and generate them simultaneously. This limits their ability to adapt to new queries that arrive in a streaming fashion, where future queries are fundamentally unknown.

An appealing solution is therefore to adopt a causal formulation of NVS, where views are generated autoregressively, each conditioned only on previously observed inputs and synthesized outputs. However, introducing causality into multi-view diffusion models poses two key challenges: (1) *Autoregressive Drift.* This arises from the mismatch between training and inference. The model learns from clean ground-truth views during training but must rely on its own imperfect autoregressive outputs at inference time, causing errors to accumulate and degrade future predictions. (2) *KV Caching Incompatibility.* A key advantage of autoregressive models lies in efficient inference through KV caching (Lages, 2023), which enables the reuse of attention computations from previously generated views. However, most existing NVS models (Watson et al., 2024; Zhou et al., 2025b) rely on absolute pose encodings, such as Plücker rays (Sitzmann et al., 2019), that are tied to a global coordinate system. When the reference frame shifts (i.e., the global coordinate changes), the cached attention results become invalid and must be recomputed. Alternatively, keeping the reference frame fixed avoids recomputation, but over long rollouts, the absolute pose values grow progressively larger, eventually drifting out of training distribution, leading to a noticeable drop in generation quality.

Building on these insights, we propose CausNVS, an autoregressive multi-view diffusion model for flexible and scalable novel view synthesis. The core idea is to introduce a framewise attention layer with causal masking on top of a

---

[1]Anonymous Institution, Anonymous City, Anonymous Region, Anonymous Country. Correspondence to: Anonymous Author <anon.email@domain.com>.

Preliminary work. Under review by the International Conference on Machine Learning (ICML). Do not distribute.

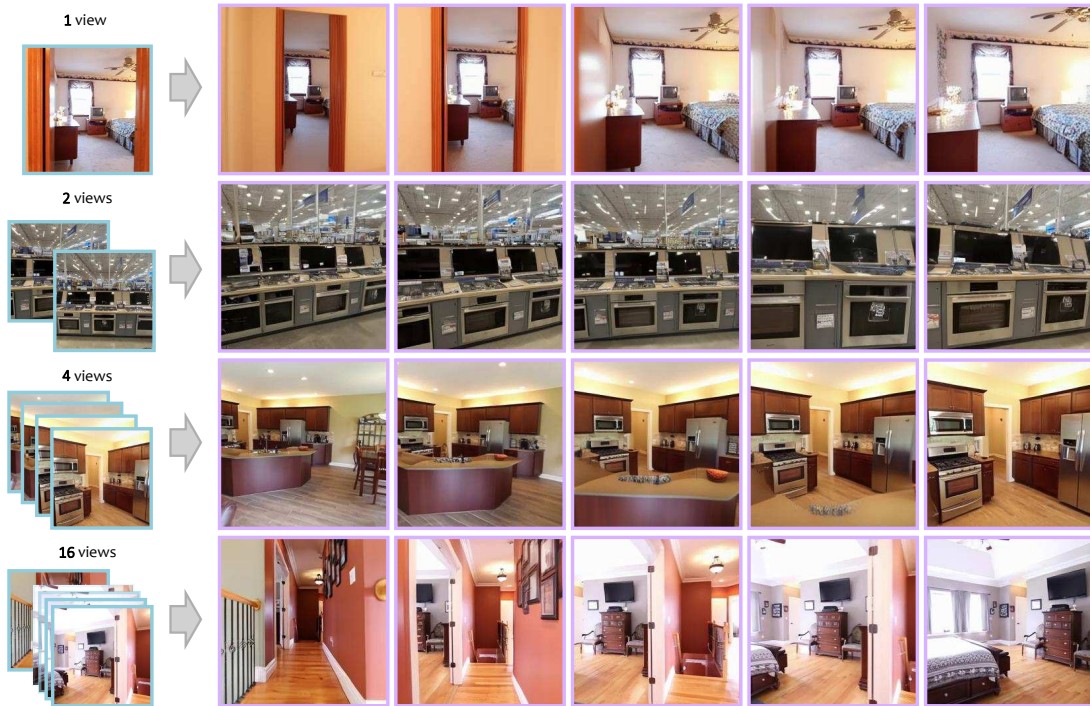

*Figure 1.* **Autoregressive Novel View Generation.** CausNVS generates target views autoregressively from an arbitrary number of input views. Albeit trained with sequences of 8 frames, it is capable of longer rollouts maintaining 3D consistency, and results continue to improve with more input views.

pretrained 2D diffusion backbone. Unlike non-causal models, our formulation enables autoregressive generation and inherently supports arbitrary input-output view configuration within a single training pass, substantially improving training efficiency.

To mitigate autoregressive drift, we apply per-frame noise conditioning during training (Chen et al., 2024a), which encourages the model to learn from uncertain or imperfect contexts rather than relying solely on clean ground-truth inputs. At inference time, this enables noise conditioning augmentation (Ho et al., 2022; Valevski et al., 2025), where we can assign a small noise level to previously generated views, allowing the model to treat them as noisy inputs. This stabilizes subsequent predictions and improves robustness to accumulated errors. To enable efficient autoregressive inference with KV caching, we adopt CaPE (Kong et al., 2024), a relative pose encoding that captures pairwise camera relationships independently of global coordinates. This design ensures that cached attention remains valid as the viewing window shifts, allowing for efficient sliding-window inference without recomputation or cache invalidation.

Extensive evaluation shows that CausNVS delivers strong visual quality on fixed-view benchmarks, surpasses SoTA large-scale baselines in flexible NVS, and supports stable rollouts up to 10× longer than training, demonstrating strong

generalization for real-time and open-ended synthesis.

## 2. Related Work

**Multi-view Diffusion Models**    Most multi-view diffusion models (Gao* et al., 2024; Wu et al., 2025; Watson et al., 2024; Wang et al., 2024b; Tseng et al., 2023; Yu et al., 2024a; 2023) operate under a fixed-view setting, where the total number of views is fixed and typically consists of one or two input views along with a predefined set of target cameras. All target views are jointly denoised to enforce cross-view consistency. Some methods (Sun et al., 2024; Yu et al., 2024b) further enhance geometric consistency by incorporating explicit scene geometry, such as depth maps or point cloud projections, as additional guidance. These approaches usually require extra geometry estimation pipelines (Wang et al., 2024a) and tend to degrade when there are large pose differences between input and target views. Another line of work enhances 3D representations by leveraging pretrained image or video priors (Liu et al., 2024; Tang et al., 2023; Poole et al., 2022), for instance by refining NeRF (Mildenhall et al., 2020) or 3D Gaussian Splatting (Kerbl et al., 2023). These approaches typically rely on computationally expensive iterative optimization. To support fast inference, recent feedforward models (Chen et al., 2024b;c; Xu et al., 2025; Jin et al., 2025) directly

regress novel views using transformers, but they often struggle to generalize to sparse or distant viewpoints due to the lack of generative modeling capacity.

To support more flexible view configurations, SEVA (Zhou et al., 2025b) proposes a two-stage pipeline that clusters target views, generates anchor frames, and interpolates intermediate views using two separate diffusion models. While more adaptive than fixed-view baselines, SEVA relies on full trajectory preprocessing and handles variable-length inputs via view anchoring and padding. As a result, it still operates on fixed-size input–output tuples during inference, limiting its suitability for streaming applications.

**Autoregressive Diffusion Models**  The autoregressive formulation also aligns with recent advances in temporal diffusion models, which support progressive or streaming generation. MAGI-1 (Sand-AI, 2025) applies chunk-wise autoregressive denoising to videos, enabling long-horizon synthesis with prompt control. Diffusion Forcing (Chen et al., 2024a) introduces soft noise-level masks to unify autoregressive and parallel modes, while CausVid (Yin et al., 2025) distills bidirectional video models into causal students for low-latency autoregressive rollout. Self-Forcing (Huang et al., 2025) minimizes the train-inference gap by simulating the inference process during training. However, these methods operate in the time domain and do not address geometric conditioning dependencies.

In parallel, Game-n-Gen (Valevski et al., 2025) and Genie (Parker-Holder et al., 2024; Ball et al., 2025) investigate autoregressive generation in interactive or physically grounded environments. Unlike our NVS formulation, world models (i) rely on actions that cannot be directly mapped to camera poses due to intervening physical dynamics, and (ii) assume strictly sequential temporal evolution, whereas NVS may involve non-continuous view sampling without temporal order.

**3D Spatial Memory and Camera Pose Encoding**  As autoregressive generation scales to longer sequences, two complementary capabilities become essential: spatial memory to retain *what was there* over time, and pose conditioning to synthesize *what is there* in each frame. The former supports long-horizon consistency, while the latter enables geometric alignment across views. This connects to recent progress in world models, which aim to synthesize coherent environments over extended horizons (Agarwal et al., 2025; Parker-Holder et al., 2024; Alonso et al., 2024; Valevski et al., 2025). However, many models lack explicit 3D pose grounding or struggle with maintaining spatial consistency due to limited context size. To address this, several works (Wang & Agapito, 2024; Wang* et al., 2025; Xiao et al., 2025) propose memory mechanisms for 3D-aware generation, often based on state-space models (Hochreiter & Schmidhuber, 1997) or external memory banks.

Meanwhile, pose conditioning has advanced for more generalizable view synthesis and precise camera control. Traditional absolute encodings, such as Plücker rays (Sitzmann et al., 2019) or raw extrinsics (Liu et al., 2023; Wang et al., 2024b; He et al., 2025), are tied to global coordinates and often overfit to specific view settings. EscherNet (Kong et al., 2024) addresses this by introducing a relative Camera Pose Encoding (CaPE) that enables more flexible view configurations without retraining. While CaPE introduces a more generalizable representation, integrating spatial memory with pose-aware conditioning remains underexplored. Our work extends this direction by leveraging KV caching and pose-aware sliding-window attention to form a transformer-native memory that supports spatially consistent generation in streaming and variable-view settings.

## 3. CausNVS: Causal Diffusion Model

We start with a brief formulation of the autoregressive novel view synthesis (NVS) task. Given a set of $N$ possibly discontinuous posed input views $\{(\boldsymbol{x}_1, \boldsymbol{p}_1), \ldots, (\boldsymbol{x}_N, \boldsymbol{p}_N)\}$ and a sequence of $M$ target camera poses $\{\boldsymbol{p}_{N+1}, \ldots, \boldsymbol{p}_{N+M}\}$, where $\boldsymbol{x}_i$ is an RGB image and $\boldsymbol{p}_i$ is the corresponding camera pose (extrinsics and intrinsics), the goal is to generate the corresponding target views $\{\boldsymbol{x}_{N+1}, \ldots, \boldsymbol{x}_{N+M}\}$. Unlike the non-autoregressive setting, where all target poses are specified in advance, the autoregressive setting assumes that target poses may arrive sequentially during inference, and thus future poses remain unknown. Consequently, each target view $\boldsymbol{x}_i$ must be synthesized sequentially, conditioned only on the input views and the previously generated frames.

**Training Objective**  CausNVS formulates a latent diffusion model (Ho et al., 2020; Rombach et al., 2022) well-suited for the above problem setup as follows: Let $F = N + M$ denote the total number of frames in each training sequence. Each frame $\boldsymbol{x}_i, i = 1, ..., F$, is encoded into a latent $\boldsymbol{z}_i$ using a pretrained VAE encoder and we define the corresponding view as $\boldsymbol{v}_i = (\boldsymbol{z}_i^{t_i}, \boldsymbol{p}_i)$, where $\boldsymbol{z}_i^{t_i}$ denotes the latent $\boldsymbol{z}_i$ at noise level $t_i$ and $\boldsymbol{p}_i$ represents camera poses. Here, we sample random independent noise levels $t_i$ for each frame and encode them. This allows the model to learn denoising conditioned on partially noisy views, reducing the training-inference gap during autoregressive generation.

Further, to ensure each frame is conditioned only on past information, causal masking is applied across spatial view tokens in frame-wise attention, similar to Diffusion Forcing (Chen et al., 2024a) but within a UNet transformer architecture rather than a recurrent network. Under this setting, the model $\hat{\boldsymbol{\epsilon}}_\theta$ is trained o predict the noise $\boldsymbol{\epsilon}_i$ added to each latent $\boldsymbol{z}_i$, conditioned on its causal context $\boldsymbol{v}_{<i}$. Mathematically, the training objective is:

$$\mathcal{L} = \mathbb{E}_{\{(\boldsymbol{x}_i, \boldsymbol{p}_i), t_i, \boldsymbol{\epsilon}_i\}_{i=1}^F} \sum_{i=1}^F \|\hat{\boldsymbol{\epsilon}}_\theta(\boldsymbol{v}_i | \boldsymbol{v}_{<i}) - \boldsymbol{\epsilon}_i\|_2^2 \quad (1)$$

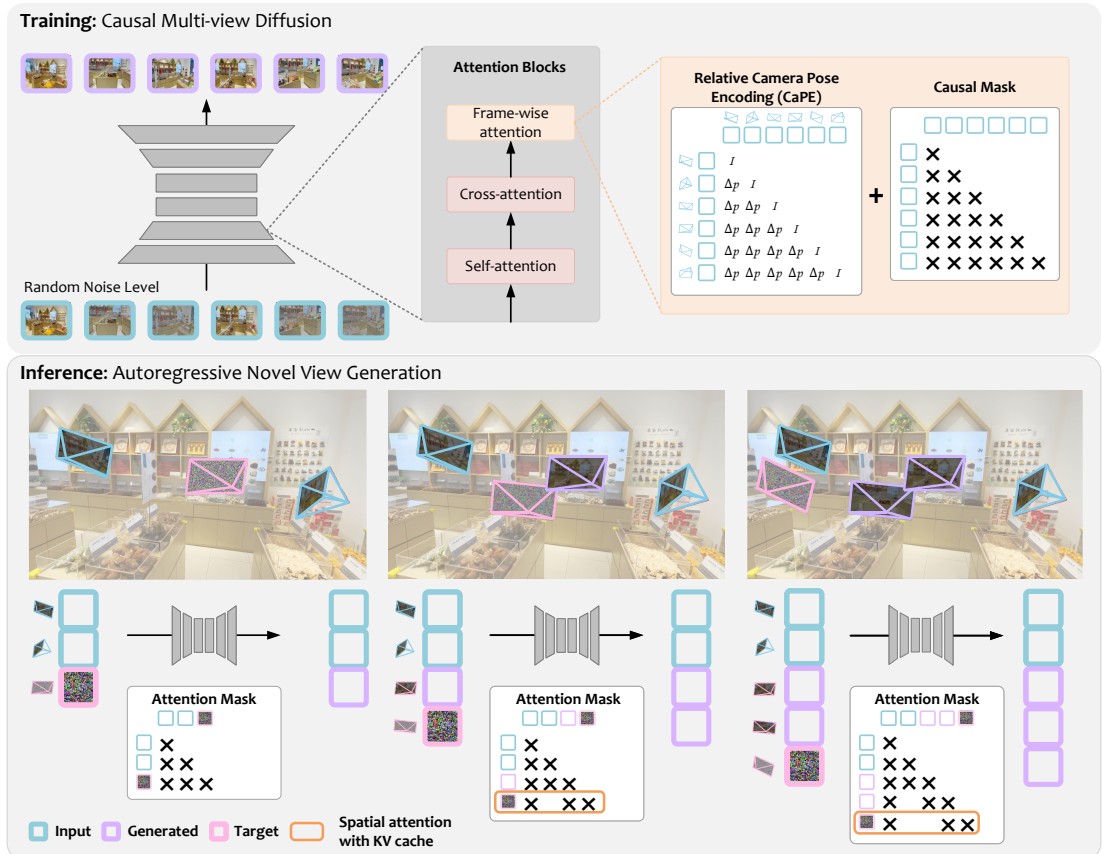

*Figure 2.* **Causal Multi-view Diffusion Pipeline.** Each view is tokenized with its camera pose and noise level, and processed with frame-wise self-attention, causal masking, and CaPE. At inference time, given a variable number of conditioning views, the model performs autoregressive denoising using KV caching with spatial attention window.

where $\epsilon_i \sim \mathcal{N}(0, I)$. This setup offers a key advantage: the model effectively learns a wide range of input-target view configurations within a single training pass thanks to teacher forcing (Williams & Zipser, 1989). Consequently, it generalizes not only to arbitrary $N$-to-$M$ view predictions without requiring separate training or padding, similarly to contemporary language models, but also to varying levels of noise conditioning due to training with per-frame noise. In contrast to prior approaches (Watson et al., 2024; Zhou et al., 2025b; Gao* et al., 2024; Sargent et al., 2024), which typically apply bidirectional denoising with shared noise levels across target views, our causal formulation with independently sampled noise allows more flexible conditioning and autoregressive generation. This also mitigates trajectory-induced biases (e.g., in 4DiM (Watson et al., 2024), roads tend to align with the camera path) and reduces over-reliance on the overall trajectory shape.

**Noise-Level Conditioning** In contrast to prior video or multi-view diffusion models (Blattmann et al., 2023; Watson et al., 2024; Zhou et al., 2025b) that apply a *uniform* noise level across all frames, our *independent* frame-wise

noise ensures that, at inference time, conditioning on previous views with lower noise remains in-distribution even if the current frame is sampled at a higher noise level. To further improve robustness during long rollouts, we can optionally apply noise conditioning augmentation by setting small noise levels for previously generated frames, which reflects their uncertainty and helps model reweight context and correct flawed predictions. Moreover, unlike sequential scheduled noise in Diffusion Forcing (Chen et al., 2024a), which assumes continuous temporal or viewpoint ordering, our setup supports arbitrary query pose orderings, a key requirement for novel view synthesis where target views may not follow a regular or continuous trajectory.

**Relative Camera Pose Encoding (CaPE)** To leverage large-scale pre-trained image priors, we finetune a text-to-image UNet diffusion backbone similar to LDM (Rombach et al., 2022) by inserting additional frame-wise attention layers into each attention block, as illustrated in Fig. 2. In these layers, we apply CaPE (Kong et al., 2024) to the query and key projections, encoding view-to-view geometry in a coordinate-invariant and scalable manner. Unlike *absolute*

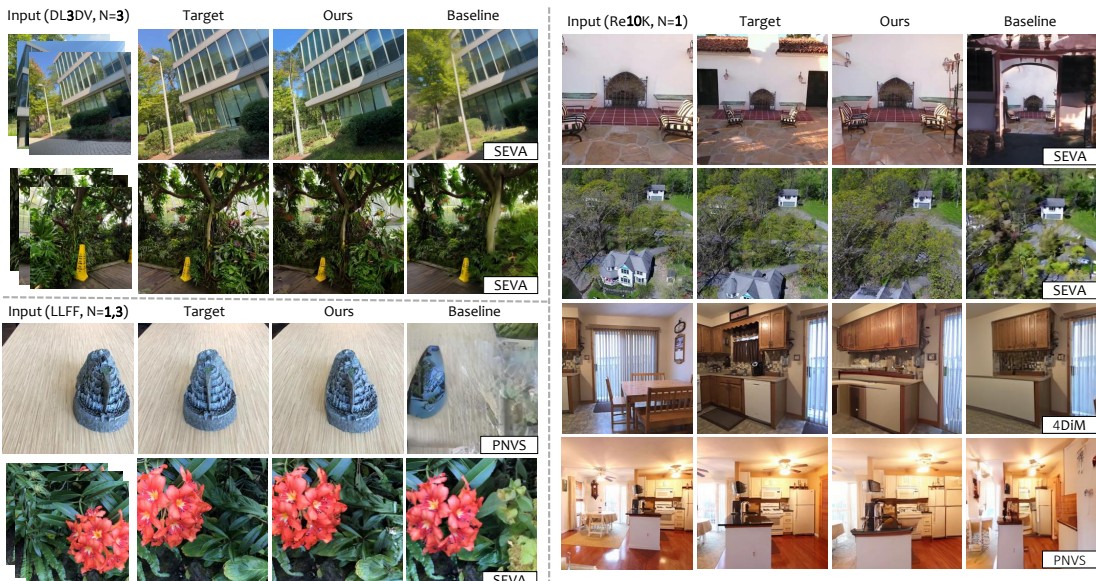

*Figure 3.* **Novel View Synthesis Results.** Visual comparison of our method and baseline models across diverse scenes and camera trajectories with input images from the DL3DV, RealEstate10K, and LLFF datasets. Each example shows the input views ($N$), the ground-truth target, our prediction, and a baseline result. While all baselines generate target views concurrently, ours generates autoregressively.

*Table 1.* **Novel View Synthesis Benchmark (PSNR↑).** Most baselines are designed for single-view or fixed-view conditioning and generate all target views concurrently. In contrast, our model, trained with a fixed sequence length $F = 8$, supports flexible input configurations and autoregressive generation. Despite the shorter training horizon, it achieves competitive performance across diverse datasets and view counts, while maintaining stable quality over long generation rollouts. Note: SEVA results on RealEstate10K are obtained using the official open-source implementation.

| Method | Dataset | RealEstate10K | | | LLFF | | DL3DV – Long (Short) | | |
|---|---|---|---|---|---|---|---|---|---|
| | $N$ | 1 | 2 | 4 | 1 | 3 | 1 | 3 | 6 |
| **Feed-forward** | | | | | | | | | |
| MVSplat | | 20.42 | – | – | 11.23 | 12.50 | – | – | 14.34 |
| DepthSplat | | 20.90 | – | – | 12.07 | 12.62 | 9.63 | 12.52 | 15.72 |
| **Generative** | | | | | | | | | |
| PNVS | | 16.07 | – | – | 12.04 | – | – | – | – |
| MotionCtrl | | 12.74 | – | – | 9.72 | – | – | – | – |
| 4DiM | | 17.08 | – | – | 11.58 | – | – | – | – |
| ViewCrafter | | 20.43 | – | – | 10.53 | 13.52 | 8.97 | 11.50 | 13.78 |
| SEVA | | 12.37 | 16.71 | 19.70 | 14.03 | 19.48 | 13.01 | 15.95 | 17.80 |
| **CausNVS** | | 16.94 | 20.90 | 21.22 | 16.53 | 16.85 | 12.86 (13.29) | 14.42 (18.32) | 15.22 (18.75) |

pose encodings, such as Plücker rays (Sitzmann et al., 2019; Gao* et al., 2024; Watson et al., 2024) and extrinsics injection (Liu et al., 2023; He et al., 2025), which are tied to global coordinates and introduce extra learnable parameters, CaPE encodes only *relative* pose relationships, defined as:

$$\pi(\mathbf{v}, \mathbf{P}) = (\mathbf{I}_{d/4} \otimes \Psi)\mathbf{v}, \quad \Psi = \begin{cases} \mathbf{P} & \text{if key} \\ \mathbf{P}^{-\top} & \text{if query} \end{cases} \quad (2)$$

where $\mathbf{v} \in \mathbb{R}^d$ is the feature vector of a view, serving as a query or key token in attention, and $\mathbf{P}$ is the corresponding camera pose. $(\mathbf{I}_{d/4} \otimes \Psi) \in \mathbb{R}^{d \times d}$ is a block-diagonal matrix constructed by tiling the $4 \times 4$ camera pose matrix $\Psi$. Similar to RoPE (Su et al., 2021), CaPE operates by rotating the query and key vectors in the attention computation without adding any parameters. This formulation ensures that attention scores are invariant to global coordinate changes, which is critical for causal attention and KV

caching reuse in flexible NVS. In contrast, absolute pose encodings can only reuse the cache if the reference frame is preserved throughout generation; dropping it or shifting the reference would alter all subsequent pose encodings and require recomputation. As trajectories become longer, the corresponding absolute coordinates also grow larger and often exceed the normalized bounds seen during training, leading to out-of-distribution behavior. By encoding only relative pose relationships, CaPE eliminates this dependency on a fixed reference frame, allowing frames to be dropped or cached in a sliding-window and enabling scalable autoregressive generation with arbitrary query orderings. Additionally, CaPE yields cyclic variation under $360°$ rotations and linear dependence on translations, detailed in Fig. 6.

**KV Caching as Spatial Memory** To enable scalable inference across long view sequences, we adopt a lightweight

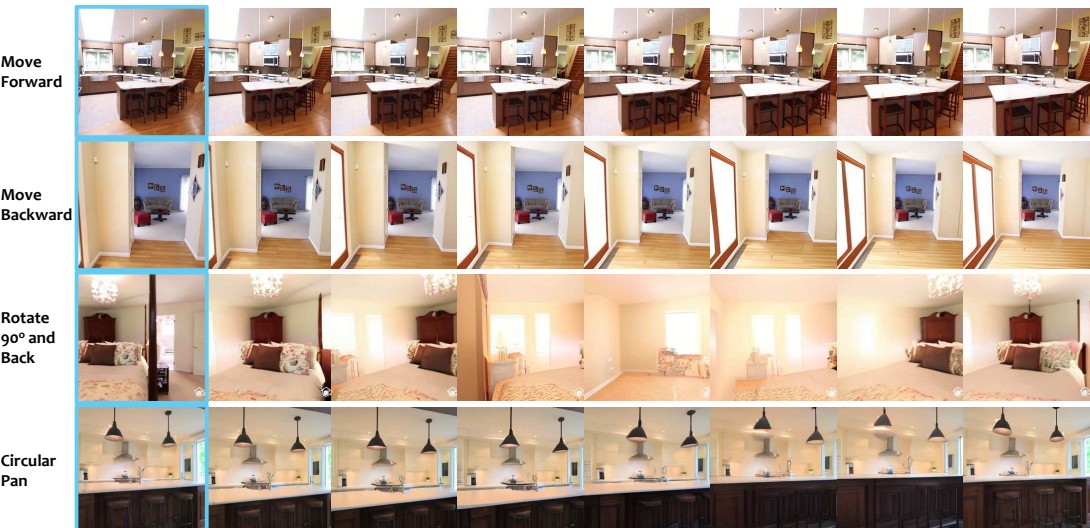

*Figure 4.* **Novel View Synthesis on Diverse Customized Trajectories.** CausNVS generalizes to diverse camera motions, including trajectories that return back, showcasing the spatial consistency.

KV caching (Lages, 2023) design tailored for diffusion models. A key distinction in our diffusion settings is that KV caching is required only in frame-wise attention layers, and writing to the KV cache is only necessary in the last denoising step (or the noise-augmented conditioning step), rather than across all denoising steps. This is enabled by our per-frame noise training, which supports conditioning at arbitrary noise levels during inference. In contrast, traditional video diffusion models use bidirectional attention and jointly denoise target frames, making KV caching incompatible. To further reduce attention overhead and preserve relevance, we apply a pose-aware sliding window that restricts attention to the top-$K$ nearest views in pose space. This is possible because we train with randomized view sets, ensuring the model learns to operate under arbitrary view orderings. Together, these choices yield an implicit and efficient 3D spatial memory: each generated view can access the most geometrically relevant prior content without recomputation. Unlike external memory systems (Wang* et al., 2025; Xiao et al., 2025) that rely on separate memory buffers and explicit, lossy updates over long sequences, our cache is integrated into transformer with minimal overhead.

## 4. Experiments

We train CausNVS on two public scene-level datasets with camera pose annotations: RealEstate10K (Zhou et al., 2018) and DL3DV (Ling et al., 2024). For each training instance, a sequence of $F=8$ frames is randomly sampled from a single scene, where the selected frames may be non-contiguous. Our model contains 915M parameters, and is finetuned from a pre-trained latent diffusion model, at resolution $256 \times 256$ with batch size 128, on 16 v5e TPUs, detailed in Appendix A.

We evaluate our method on novel view synthesis tasks across a wide range of flexible input views $N \in \{1, 2, 3, 4, 6\}$ and output views $M \in [1, 80]$, demonstrating the ability of our model to generalize across arbitrary $N$-to-$M$ synthesis settings with a single trained model. We conduct evaluation following the public SEVA benchmark (Zhou et al., 2025b) settings across Re10K (Zhou et al., 2018), LLFF (Mildenhall et al., 2019), and DL3DV (Ling et al., 2024) datasets, with various number of views. We compare CausNVS to strong baselines, including generative models: SEVA (Zhou et al., 2025b), MotionCtrl (Wang et al., 2024b), 4DiM (Watson et al., 2024), ViewCrafter (Yu et al., 2024b), PNVS (Yu et al., 2023), and feed-forward models: MVSplat (Chen et al., 2024b), DepthSplat (Xu et al., 2025). Among them, SEVA (1.5B) is first trained with a batch size of 1472 for $F=8$, and then further trained with a batch size of 512 for $F=21$, using a diverse mixture of object- and scene-level datasets at significantly larger scale. 4DiM is jointly trained on 3D scenes and 30 million video samples, benefiting from additional temporal supervision. ViewCrafter incorporates explicit point cloud reprojection as pose guidance, supported by a 3D foundation model (Wang et al., 2024a). In contrast, our model is trained with only 20K scene-level samples at a much smaller scale, while still supporting flexible $N$-to-$M$ view synthesis with strong performance.

Further, in Sec. 4.2, we ablate the autoregressive component of CausNVS against non-autoregressive variants, highlighting the advantages of our framework in flexible N-to-M novel view synthesis with unified single-model training, as well as the efficiency and 3D consistency under novel trajectories enabled by spatial attention windows with KV caching as spatial memory.

*Table 2.* $N$-to-$M$ **Novel View Synthesis Ablation (PSNR↑ / TSED↑).** AR denotes autoregressive inference, and Parallel denotes concurrent inference. We evaluate causal and non-causal models with varying numbers of input views $N$ and total frames $F$ on the Re10K validation set. The causal model maintains consistent performance across different configurations with stable autoregressive rollout. In contrast, the non-causal model performs poorly when the evaluation sequence is shorter than its training setup, even under parallel denoising, and quickly drifts when used autoregressively.

| Model | Infer | $N$ | $F = 2$ | $F = 4$ | $F = 8$ | $F = 32$ | $F = 64$ |
|---|---|---|---|---|---|---|---|
| **Causal** | AR | 1 | 16.56 / – | 17.25 / 81.25 | 18.08 / 95.83 | 17.44 / 99.79 | 16.98 / 99.38 |
| | | 2 | – | 19.52 / 93.75 | 20.40 / 93.75 | 21.10 / 99.78 | 19.31 / 99.49 |
| | | 4 | – | – | 23.27 / 91.67 | 24.55 / 100.0 | 23.35 / 99.89 |
| | | 16 | – | – | – | 28.13 / 100.0 | 27.64 / 99.87 |
| **Non-Causal** | Parallel | 1 | 9.84 / – | 13.15 / 34.38 | 17.52 / 92.71 | 19.27 / 100.0 | 18.15 / 100.0 |
| | | 2 | – | 18.38 / 53.33 | 20.61 / 97.50 | 21.86 / 100.0 | 21.42 / 99.90 |
| | | 4 | – | – | 23.97 / 93.75 | 25.12 / 100.0 | 24.29 / 100.0 |
| | | 16 | – | – | – | 28.56 / 99.17 | 27.64 / 99.47 |
| | AR | 2 | – | – | 13.62 / 86.77 | – | – |

## 4.1. Novel View Synthesis Results

**Novel View Synthesis Benchmark** As shown in Tab. 1 and Fig. 3, CausNVS achieves competitive performance across diverse scenes and improves consistently with more input views in the SEVA benchmark (Zhou et al., 2025b). CausNVS generates views autoregressively and supports flexible $N$-to-$M$ synthesis within a single unified model, while all other baselines need to generate novel views simultaneously with fixed view counts. CausNVS maintains consistent visual quality across varying inputs and wide pose differences, where baseline results often exhibit blur or distortion. More qualitative and quantitative results are included in Appendix B.

Feed-forward methods MVSplat (Chen et al., 2024b) and DepthSplat (Xu et al., 2025) perform well when the input views are densely sampled with small baselines, but their performance deteriorates significantly under wide view gaps. Due to the lack of generative capability, they typically fail to extrapolate novel views beyond the observed ones. Generative models (Yu et al., 2023; Wang et al., 2024b; Watson et al., 2024; Yu et al., 2024b; Zhou et al., 2025b) often assume a fixed single-view input and struggle with variable-length configurations. Methods like 4DiM (Watson et al., 2024) address this limitation via multi-view fine-tuning, but at the cost of additional training overhead. ViewCrafter (Yu et al., 2024b) incorporates an explicit 3D point cloud estimator as geometric guidance, yet its performance degrades when the target view is far from the inputs. SEVA (Zhou et al., 2025b), developed concurrently with our work, supports varying numbers of input views but still assumes fixed-size input-output pairs. It relies on ad hoc heuristics such as view anchoring, chunking, and padding, combined with multiple forward passes, to approximate variable-length conditioning. In contrast, our causal model supports autoregressive generation with arbitrary numbers of input views, without requiring retraining, model modifications, or complex sampling strategies during inference.

**Single View Scale Ambiguity** Scale ambiguity remains a fundamental challenge in single-view novel view synthesis. Some baselines address this via scale sweeping: 4DiM uses pre-processed depth estimates to recover metric scale, while SEVA normalizes camera scale by sweeping from 0.1 to 2.0 and selecting the best-performing result per scene. For example, SEVA's PSNR on Re10K improves from 12.37 to 17.99 when using sweeping. However, we intentionally avoid such scale tuning during inference. First, sweeping requires access to ground truth to select the optimal scale, which limits generalization and realism. Second, our method is designed for flexible $N$-to-$M$ view synthesis in generative 3D settings, where single-view is an ambiguous special case rather than the primary focus. Third, we prioritize keeping the inference pipeline simple and consistent across varying input conditions, without hand-tuned parameters. Despite not relying on scale sweeping, our model performs competitively in the single-view setting and consistently improves as more views are provided, effectively resolving scale ambiguity via multi-view cues.

**Autoregressive Generation Drift** To evaluate the stability of our autoregressive rollout, we report results on both the short and long sequence splits of DL3DV, while keeping the input views fixed. The short sequence is a random subset of the long one, covering the same spatial region but with fewer target views. This setup enables us to isolate the effect of autoregressive drift from different input views. As shown in Tab. 1, performance on short sequences consistently improves with more input views. On long sequences, where the number of target views $M$ ranges from $4\times$ to $10\times$ the training length ($F = 8$), our model exhibits some degradation compared to the short case, reflecting the inherent challenges of long-context generation. Still, it remains competitive with most baselines. SEVA benefits significantly from its longer training sequence length ($F = 21$) and parallel decoding. We believe scaling to longer sequences would further improve long-context performance.

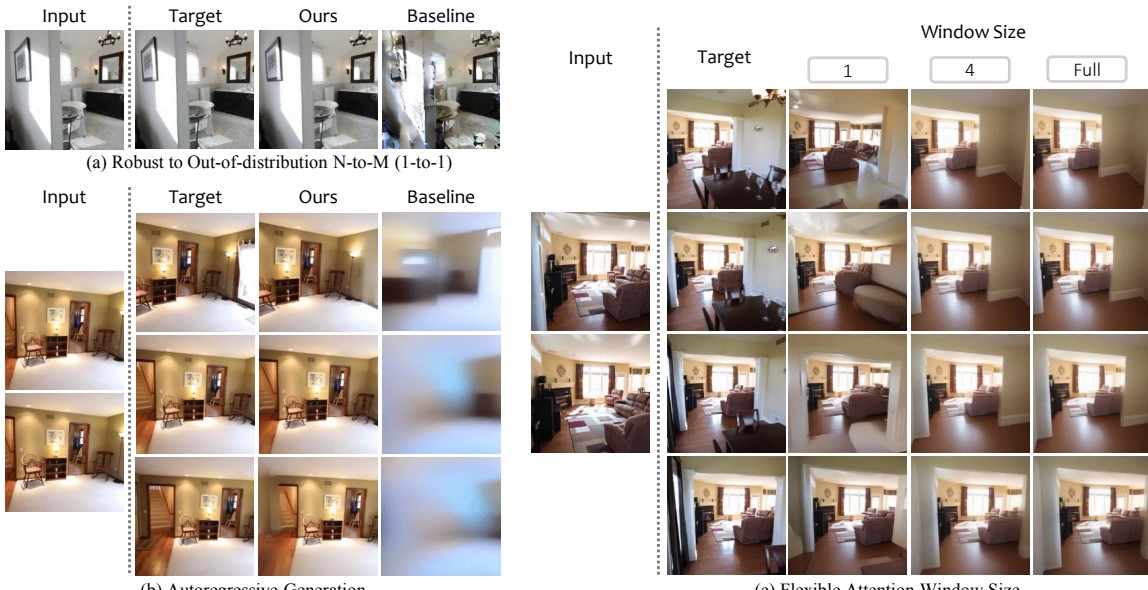

Figure 5. **Examples of Key Properties of CausNVS.** Qualitative examples highlight advantages over non-causal baselines. (a) Despite trained with a fixed sequence length $F$, causal masking enables supervision for all sequence lengths up to $F$. This leads to robust generalization, including 1-to-1 settings where non-causal models often fail. (b) Under autoregressive rollout, our model maintains visual stability, while the non-causal baseline quickly drifts and produces blurry results. (c) CausNVS supports a configurable attention window size at inference time. With window size 1, attention is limited to input views, leading to inconsistency across generated frames. With window size 4, CausNVS already achieves comparable results to full attention at a fraction of the inference FLOPS.

## 4.2. Autoregressive v.s. Non-autoregressive

$N$-**to-**$M$ **Novel View Synthesis Ablation**  We conduct an ablation study on the RealEstate10K dataset, comparing our CausNVS model to a model trained identically albeit removing causality, following established protocols (Watson et al., 2024; Zhou et al., 2025b). Both models share the same architecture and training configuration to ensure a fair comparison. We evaluate using two complementary metrics: PSNR, reflecting image quality, and TSED (Yu et al., 2023; Watson et al., 2024), a 3D structural alignment measurement between generated and ground-truth views, using a threshold of 2. As shown in Tab. 2, the causal model generalizes reliably from a single view to $8\times$ training length, and improves consistently with more input views. In comparison, the non-causal model performs well only under conditions close to its training setup and degrades significantly when evaluated with different sequence lengths. In the one-to-one generation case shown in Fig. 5, it produces unstable results and exhibits drift across frames. This is consistent with observations from previous work (Gao* et al., 2024; Kong et al., 2024), which report that non-causal models often require jointly denoising multiple padded frames to stabilize outputs. Such strategies introduce redundant computation and limit flexibility at inference time.

**Spatial Attention Window and Memory**  CausNVS supports a flexible spatial attention window that dynamically selects relevant views at inference time, enabling efficient

context aggregation without incurring full attention cost. This pose-aware sliding strategy ensures broad spatial coverage even under limited attention window, and achieves results comparable to global attention with significantly reduced FLOPS (Fig. 5c). By combining with KV caching, CausNVS retains previously generated views without recomputation, allowing context to persist implicitly across frames. We further demonstrate in Fig. 4 that this mechanism enables CausNVS to maintain spatial consistency under diverse, user-defined camera trajectories, including those with revisited views. Each generated view can retrieve geometrically relevant prior content, preserving both structure and appearance across varying and revisited viewpoints. In practice, this combination of spatial windowing and KV caching serves as an implicit and efficient spatial memory, ensuring 3D consistency.

## 5. Conclusion

We have presented CausNVS, a causal multi-view diffusion model that supports autoregressive generation with flexible numbers of input and output views. The model is designed with causal masking, noise-level conditioning, and relative pose encoding (CaPE) embedded within a unified decoder-only architecture. This design supports efficient pose-aware spatial memory and KV caching at inference time, enabling robust and efficient novel view synthesis suitable for streaming and interactive applications such as AR/VR, world modeling, and generative content creation.

**Impact Statement**   Our method may benefit applications in AR/VR, world modeling, and 3D content creation, but it also carries potential risks such as misuse in synthetic media or simulation bias. We advocate responsible model release and transparent benchmarks to support safe deployment.

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

# CausNVS: Autoregressive Multi-view Diffusion for Flexible 3D Novel View Synthesis (Appendix)

This appendix includes: (A) training and implementation details, (B) additional evaluations and visualizations of CausNVS's strengths in pose control, 3D consistency, and scalability, and (C) a discussion of limitations and statements. We highly encourage reviewers to visit our anonymous project page https://causnvs.github.io/ for further visualizations.

## A. Training Details

We finetune our CausNVS model from a latent diffusion model similar to (Rombach et al., 2022), following prior multi-view diffusion work (Gao* et al., 2024; Kong et al., 2024). The architecture adopts a UNet with 3 downsampling blocks, 1 middle block, and 3 upsampling blocks, operating in the VAE latent space where the spatial resolution is reduced from $256 \times 256$ to $32 \times 32$. To support autoregressive multi-view generation, we insert frame-wise attention layers in the attention blocks with causal masking. Due to computational constraints, these layers are added only to the deeper attention blocks at resolutions $16 \times 16$ and $8 \times 8$. These layers are zero-initialized (weights to identity, biases to zero) to preserve the pretrained prior at initialization. The model has 915M parameters and is trained for 150,000 steps with a batch size of 128 on 16 TPU v5e chips. We use the Adam optimizer with a learning rate of 1e-4 and apply a 1,000-step linear warmup. An exponential moving average (EMA) with a decay rate of 0.9999 is maintained throughout training.

## B. Additional Novel View Synthesis Results

We provide more metrics (SSIM, LPIPS) for novel view synthesis evaluation across main baselines (4DiM, SEVA) in Tab. 3.

We present more visualization:

- **Customized Pose Trajectories (Fig. 7):** CausNVS generates consistent and coherent outputs under diverse camera motion patterns such as forward/backward motion, rotations and circular pans trajectories.

- **Effect of Increasing Input Views on DL3DV (Fig. 8–11):** We observe clear benefits from more input views, yielding results that are more 3D-consistent and semantically faithful to observations.

- **Comparisons on Re10K with Baselines (Fig. 12–18):** CausNVS achieves superior pose control and scene consistency compared to strong baselines.

## C. Limitations & Statement

**Limitations and Future Directions**  While effective, our model relies on multi-step denoising (Song et al., 2020), limiting its real-time applicability. Future work could explore faster generation via consistency training (Song et al., 2023) or distillation (Yin et al., 2025; Xie et al., 2024; Sauer et al., 2024). Scaling to longer sequence and more diverse datasets (e.g., objects, videos) would improve generalization with stronger priors (Watson et al., 2024; Wu et al., 2025; Zhou et al., 2025b). Additionally, integrating audio, language, and action signals into the model could lead toward fully multimodal (Team, 2024; Zhou et al., 2025a; Xie et al., 2025; Kim et al., 2024) world models with spatial grounding and controllable rollout in complex environments.

**Reproducibility Statement**  We provide comprehensive descriptions of dataset sources, preprocessing, model architectures, hyperparameters, training schedules, inference setup, and evaluation protocols in the main text and supplementary material. All training data is publicly available, and the reported details should allow independent reproduction of our results.

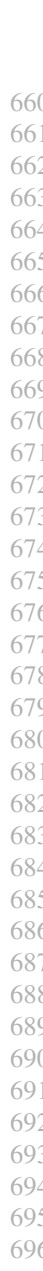

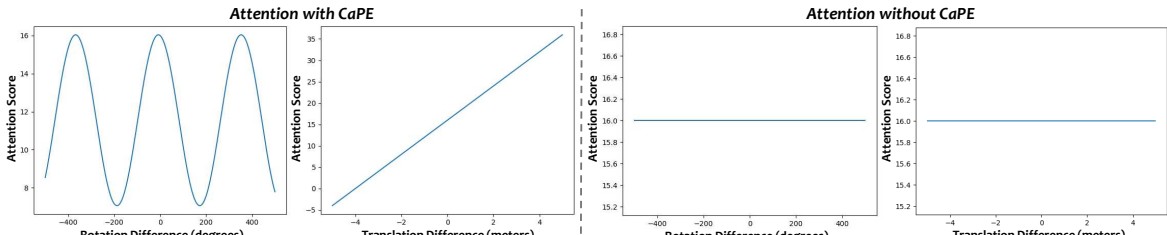

*Figure 6.* **CaPE Attention Score Analysis.** We analyze how attention responds to relative pose changes by initializing queries and keys randomly, and varying rotation or translation separately. With CaPE, attention scores change periodically with rotation and linearly with translation, indicating an SE(3)-aware inductive bias. Without CaPE, attention remains invariant to pose changes.

*Table 3.* **Novel view synthesis metrics on Re10K and LLFF.** Note: SEVA results on Re10K are obtained by running the official public code. No scale sweeping is applied at $N{=}1$.

| **Re10K** | PSNR ↑ | | | SSIM ↑ | | | LPIPS ↓ | | |
|---|---|---|---|---|---|---|---|---|---|
| $N$ | 1 | 2 | 4 | 1 | 2 | 4 | 1 | 2 | 4 |
| 4DiM | 17.08 | - | - | 0.463 | - | - | 0.302 | - | - |
| SEVA | 12.37 | 16.71 | 19.70 | 0.469 | 0.590 | 0.673 | 0.577 | 0.346 | 0.221 |
| Ours | 16.94 | 20.90 | 21.22 | 0.466 | 0.690 | 0.718 | 0.279 | 0.119 | 0.095 |

| **LLFF** | PSNR ↑ | | SSIM ↑ | | LPIPS ↓ | |
|---|---|---|---|---|---|---|
| $N$ | 1 | 3 | 1 | 3 | 1 | 3 |
| 4DiM | 11.58 | - | 0.141 | - | 0.542 | - |
| SEVA | 14.03 | 19.48 | 0.384 | 0.602 | 0.389 | 0.181 |
| Ours | 16.53 | 16.85 | 0.272 | 0.336 | 0.336 | 0.222 |

*Figure 7.* **Novel View Synthesis on Re10K with Customized Trajectories.** Results under diverse camera motion types, including trajectories that return back, showcasing our spatial consistency.

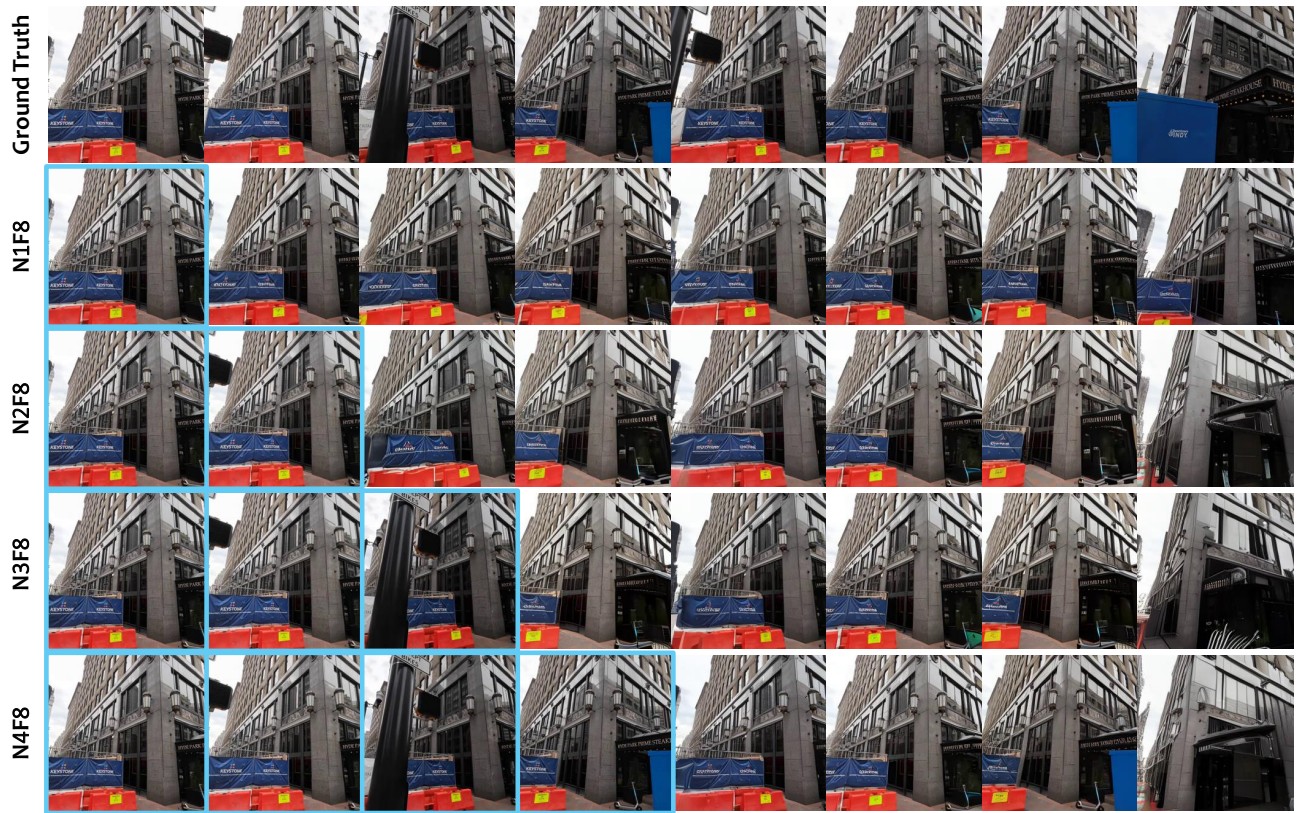

*Figure 8.* **Novel View Synthesis on DL3DV with Increasing Input Views. [Part 1]**

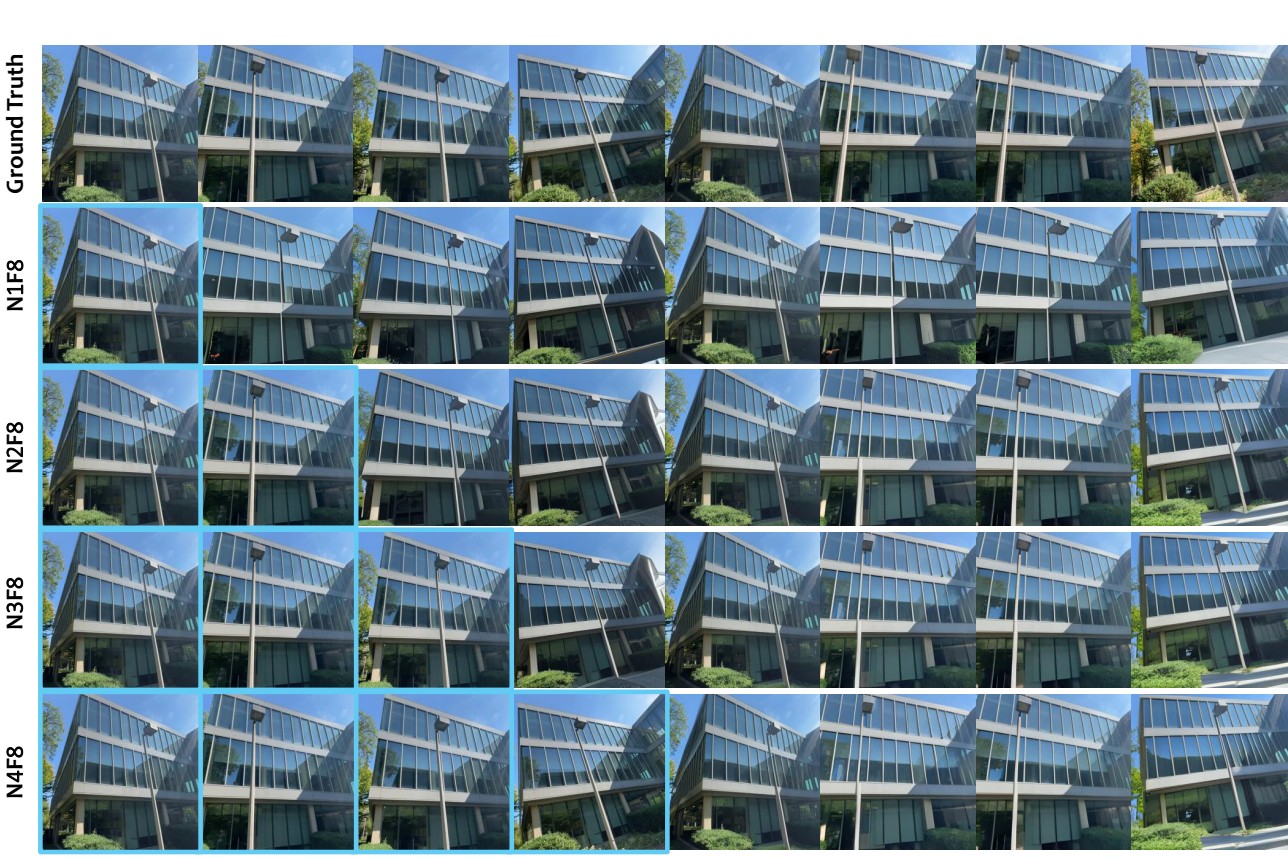

*Figure 9.* **Novel View Synthesis on DL3DV with Increasing Input Views. [Part 2]**

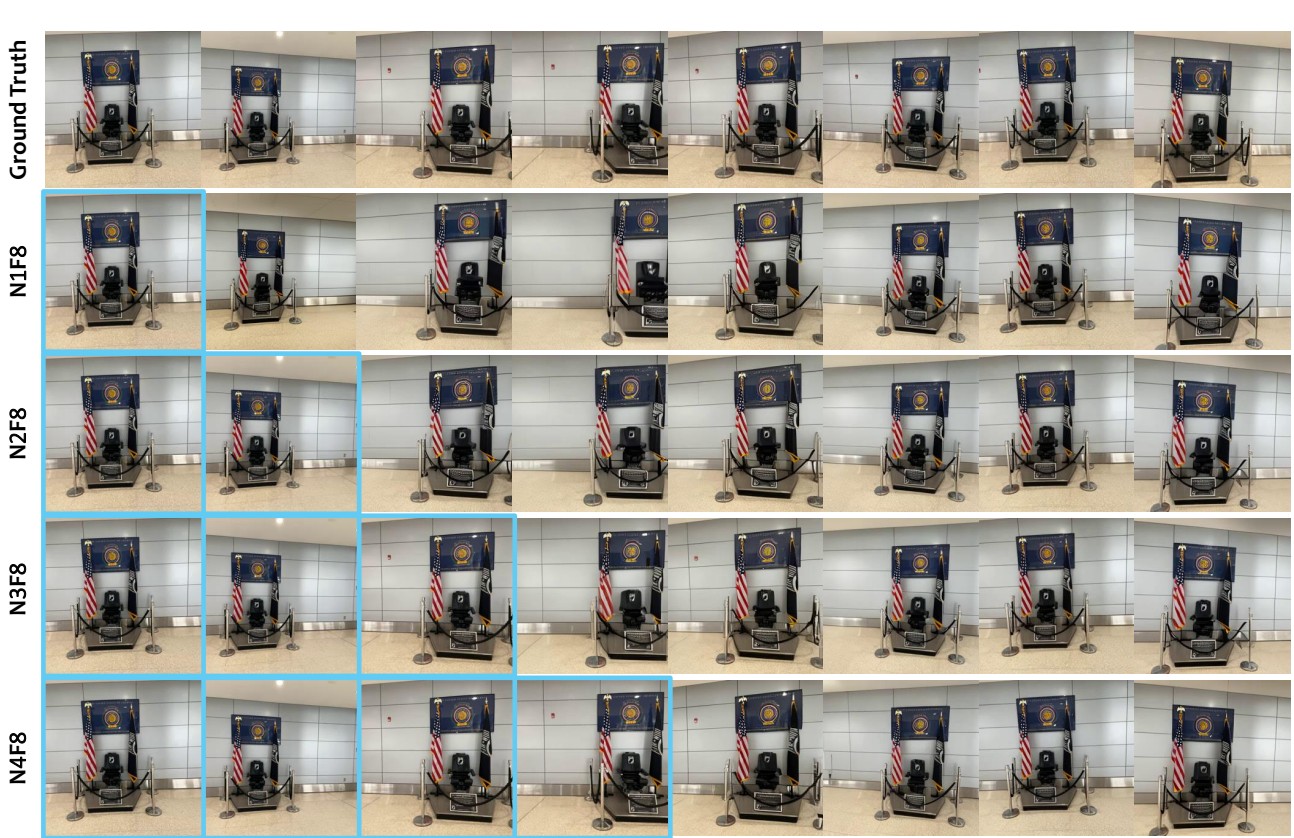

*Figure 10.* **Novel View Synthesis on DL3DV with Increasing Input Views. [Part 3]**

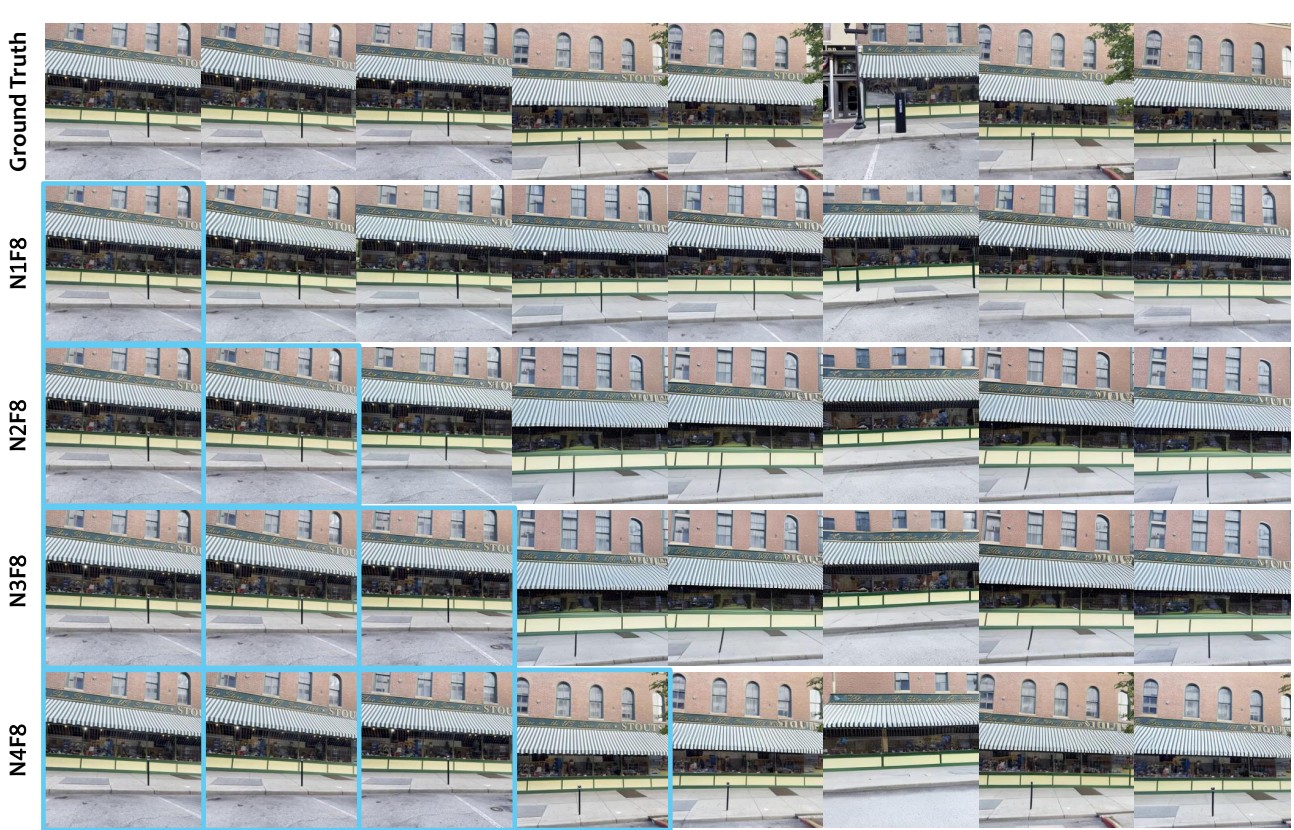

*Figure 11.* **Novel View Synthesis on DL3DV with Increasing Input Views. [Part 4]**

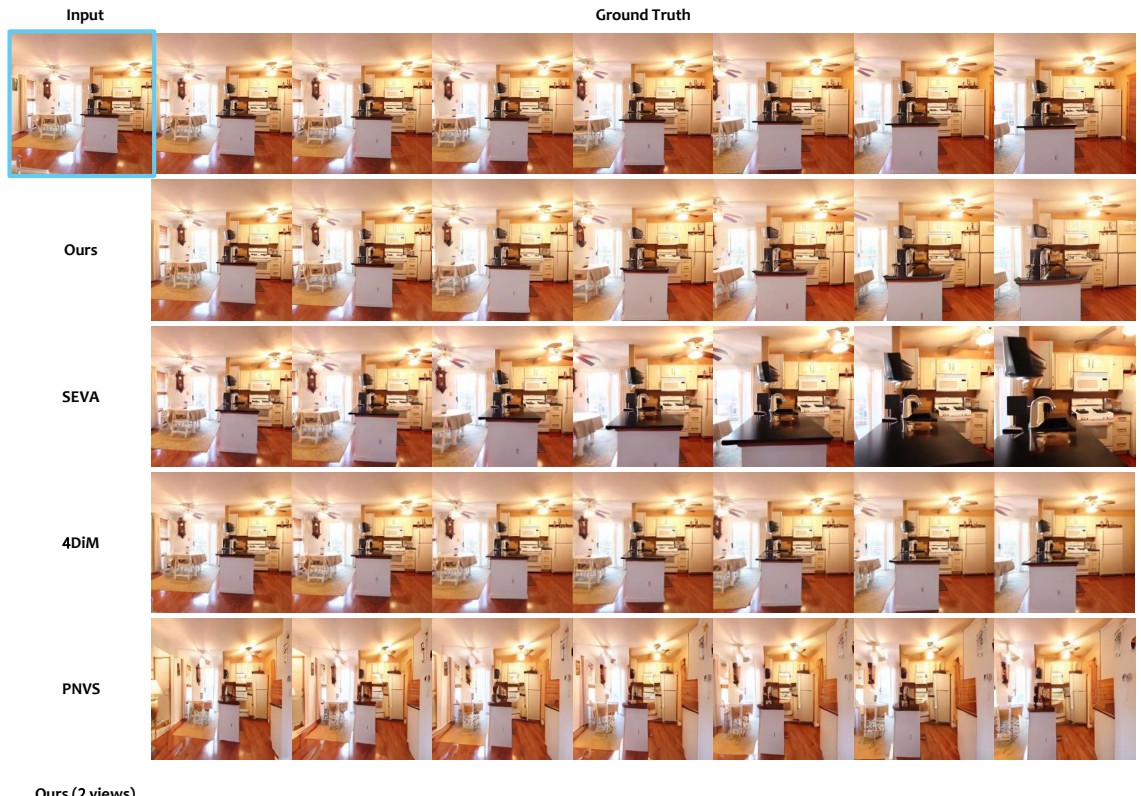

*Figure 12.* **Novel View Synthesis on Re10K Compared with Baselines. [Part 1]**

*Figure 13.* **Novel View Synthesis on Re10K Compared with Baselines. [Part 2]**

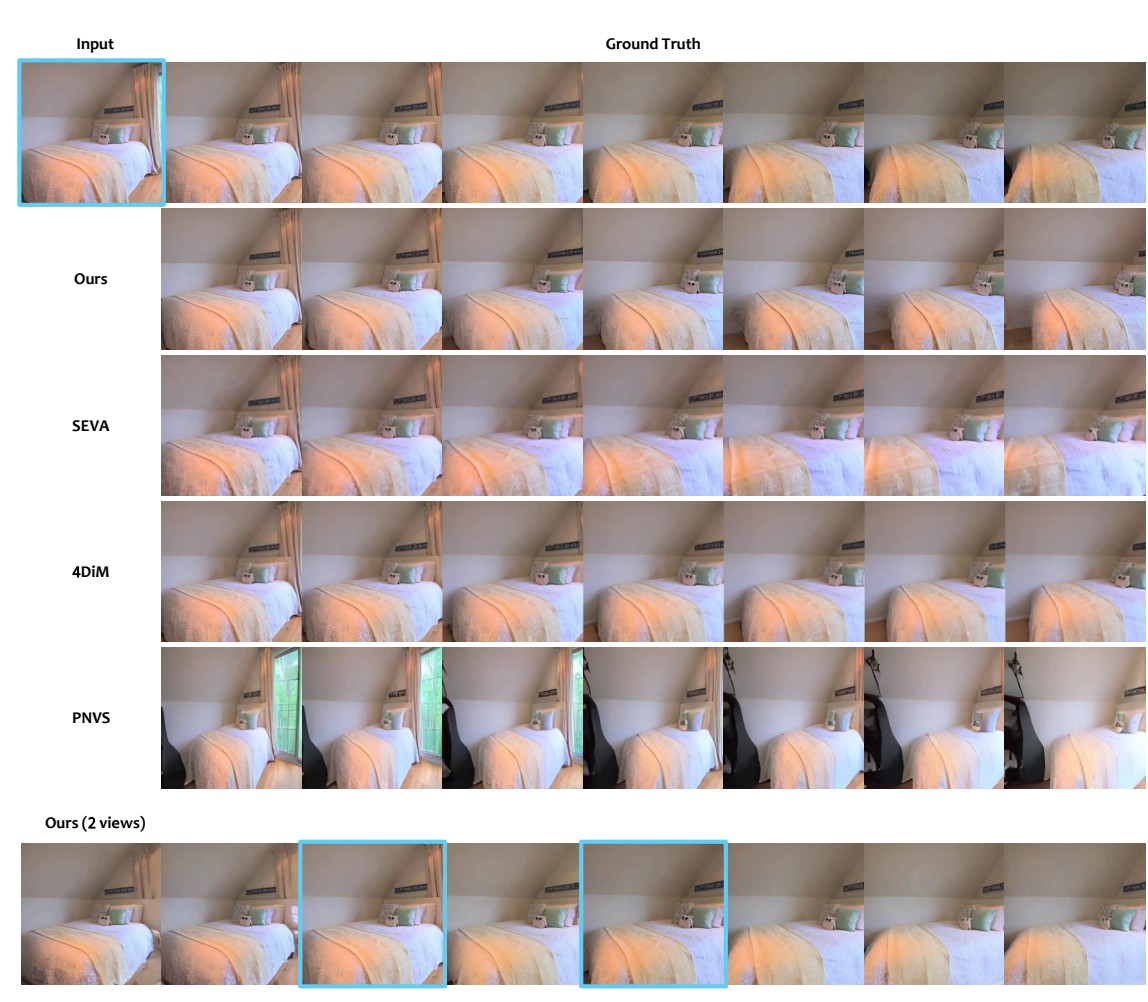

*Figure 14.* **Novel View Synthesis on Re10K Compared with Baselines. [Part 3]**

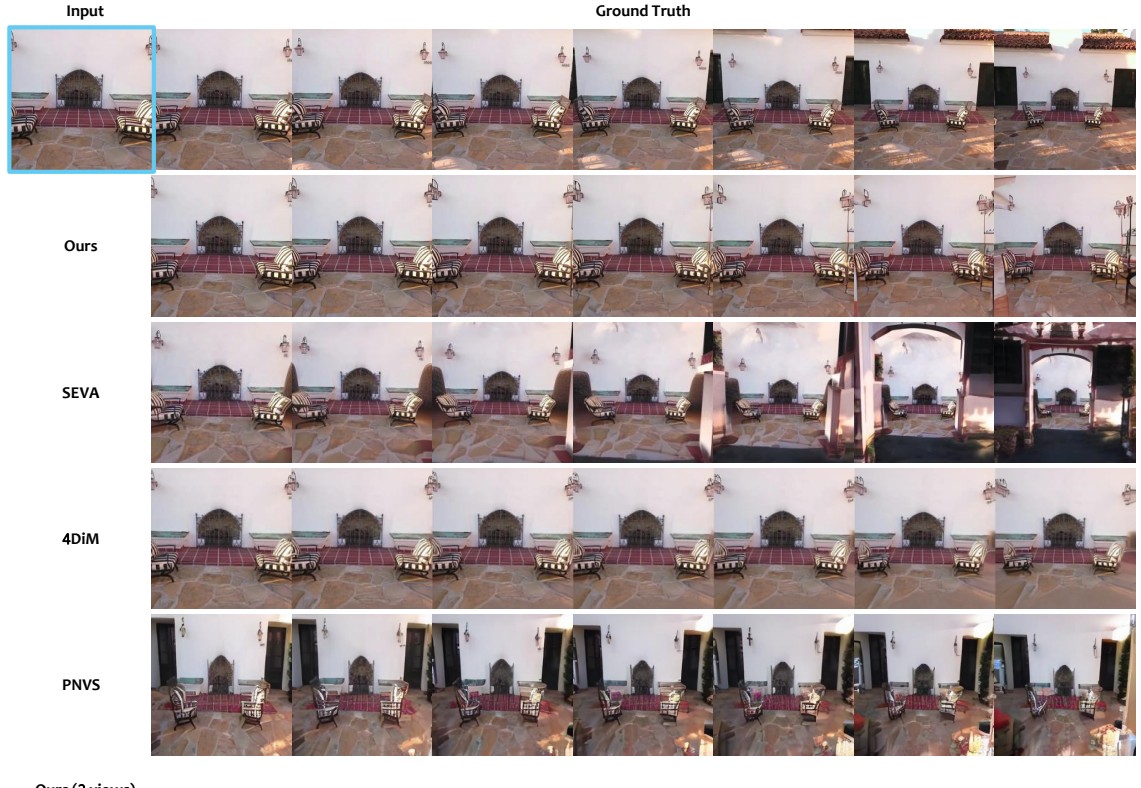

*Figure 15.* **Novel View Synthesis on Re10K Compared with Baselines. [Part 4]**

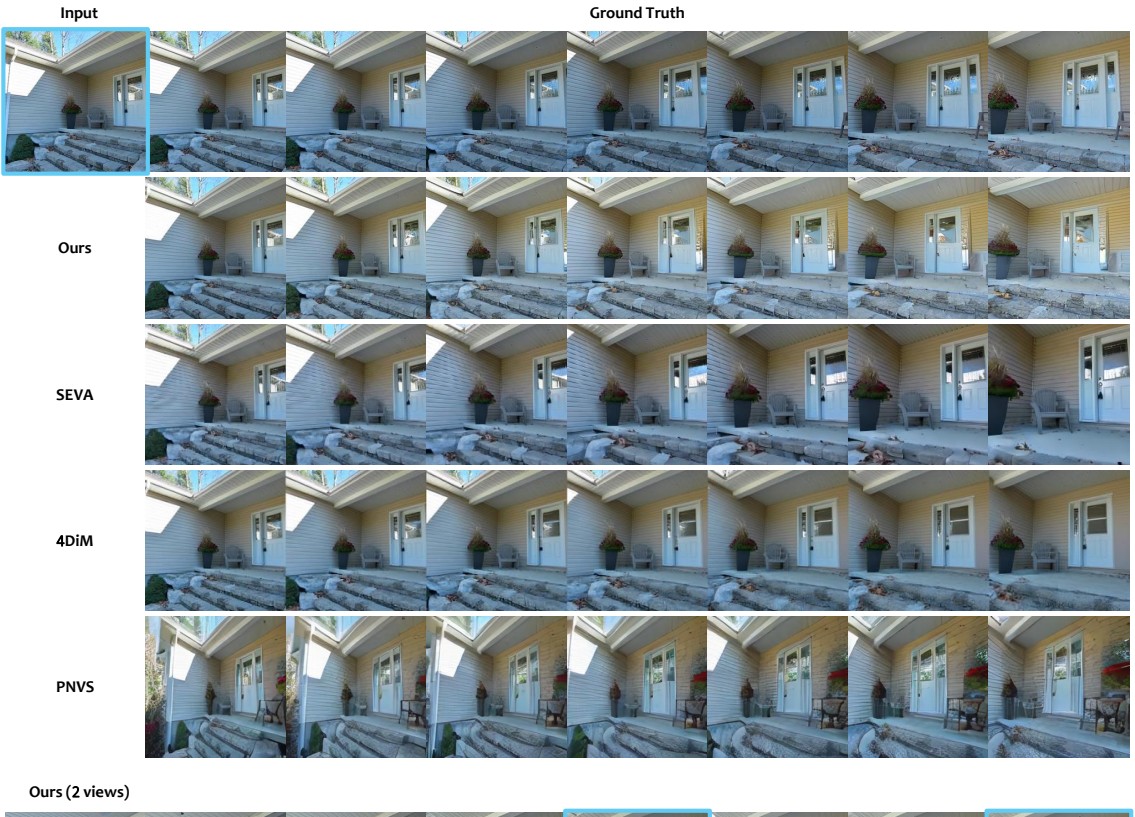

*Figure 16.* **Novel View Synthesis on Re10K Compared with Baselines. [Part 5]**

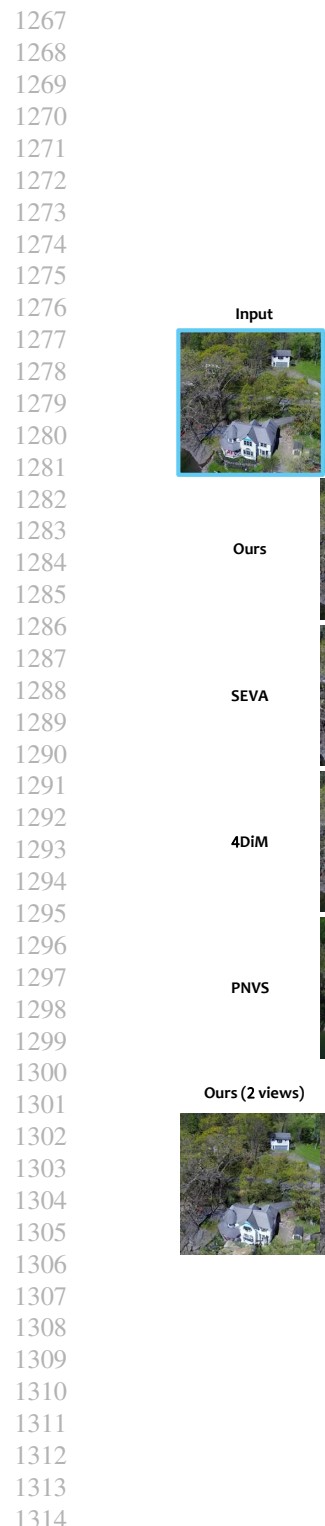

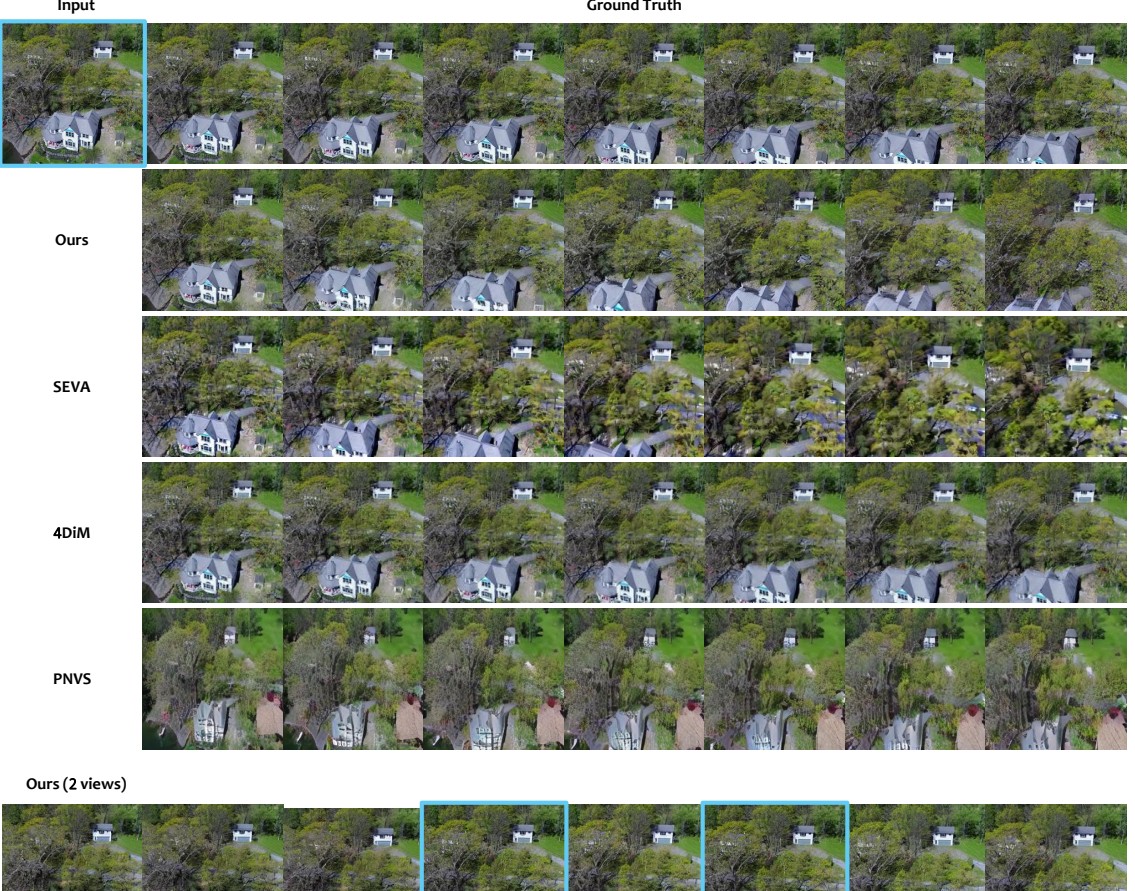

*Figure 17.* **Novel View Synthesis on Re10K Compared with Baselines. [Part 6]**

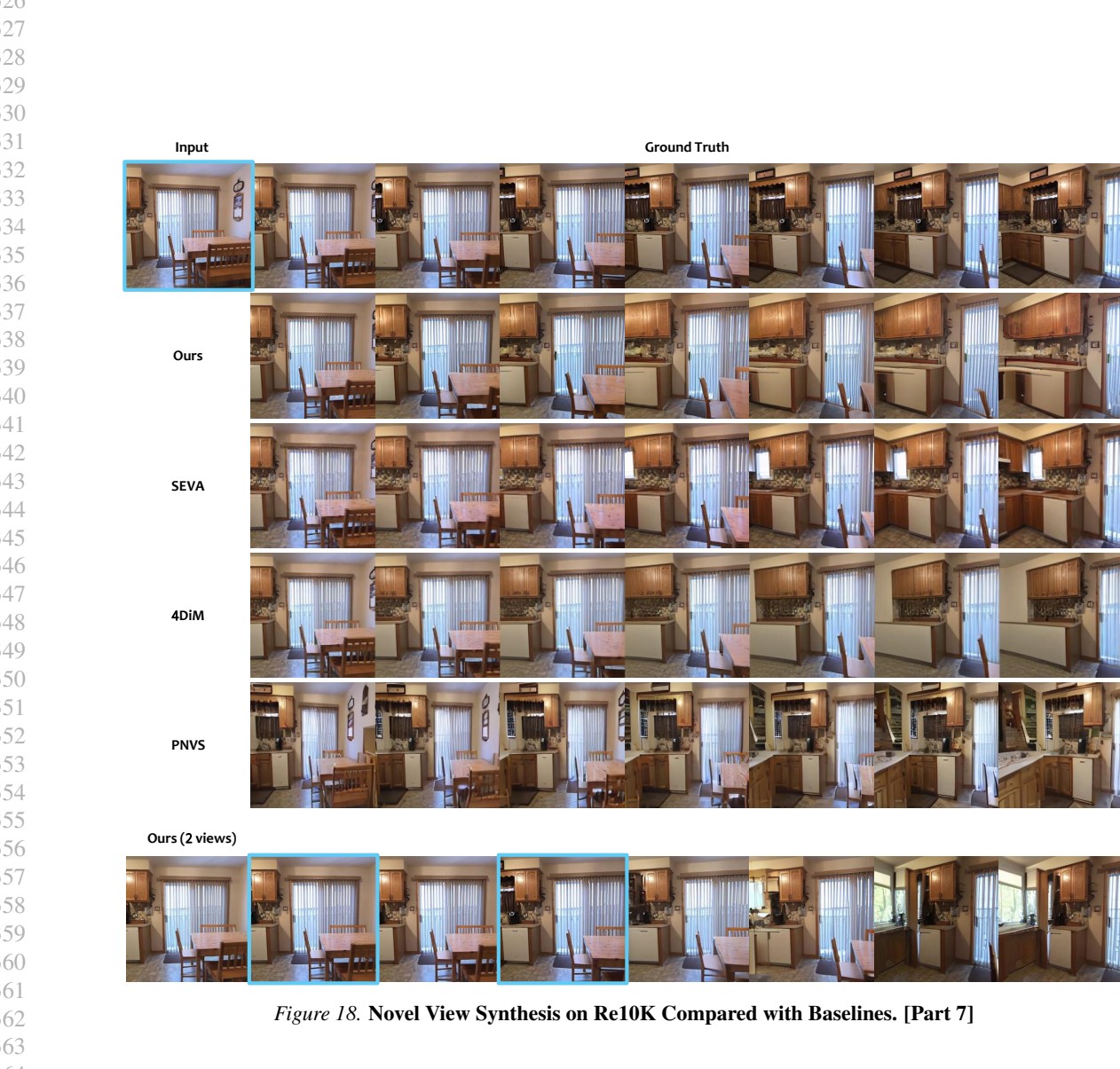

*Figure 18.* **Novel View Synthesis on Re10K Compared with Baselines. [Part 7]**

