# OpenReview forum: "CausNVS: Autoregressive Multi-view Diffusion for Flexible 3D Novel View Synthesis"
_ICML.cc/2026/Conference — Submitted to ICML 2026_

### Official Review · Reviewer_RkRu · 2026-03-09

**Soundness:** 2
**Presentation:** 3
**Significance:** 2
**Originality:** 2
**Overall Recommendation:** 4
**Confidence:** 5

**Summary:**

The method proposes a autoregressive (AR) based novel-view synthesis (NVS) method named CausNVS, utilizing modified positional embeddings, per-frame noising, and causal masking. In inference, they employ KV caching with sliding windows to eliminate drifting in rollout. Comparing to 3DGS and video generative-based NVS methods, CausNVS demonstrates more stable and flexible generation for given input sequences.

**Compliance With Llm Reviewing Policy:**

Affirmed.

**Final Justification:**

My final recommendation is 4, the rebuttal addressed the main concerns therefore I increased my score. I appreciate the responses of authors, especially clarifying about the runtime, missing baselines, and experiments about unordered camera sequences.

**Key Questions For Authors:**

The key questions for the authors are Weaknesses [2,3].

**Limitations:**

The limitations are discussed in the Appendix.

**Strengths And Weaknesses:**

Strengths:

1. The text and figures are easy to read and understand.

2. I like the N-to-M approach in general, combined with autoregression, it does make sense.

3. Supplementary material as a website is easy to read through.

Weaknesses:

1. Lack of novelty:

   - I do not think the paper ignites something in the reader for further research or gets inspired, as it already utilizes existing components: camera-relative positional embeddings from Eschernet (https://arxiv.org/abs/2402.03908) which is similar to more recently PRoPE (https://arxiv.org/abs/2507.10496) and causal masking + per-frame noise from Diffusion Forcing (https://arxiv.org/abs/2407.01392), and KV caching is standard KV caching.

2. Some important comparisons are lacking:

   - The authors compare with MotionCtrl and ViewCrafter for video models, and with MVSplat and DepthSplat for 3DGS models. If such categories are included, then I believe more recent and stronger models like GEN3C (https://research.nvidia.com/labs/toronto-ai/GEN3C/) and LVSM (https://haian-jin.github.io/projects/LVSM/) must be also included in the comparisons.

3. Autoregressive behavior is not motivated/justified adequately:

   -  There are N-to-M experiments, revealing that most compared models are overfit to specific (N,M) tuples but CausNVS generalizes better to unseen (N,M). However, why don't the authors opt for much higher M values? After all, one main reason for motivating AR can be that it could support generating much longer sequences (+100 frames) without accumulating errors, right? However, this is not demonstrated in the manuscript.

   - About smaller-scale generation for N-to-M tasks, I am confused. Because we can always take a NVS video model that supports generating K>>M frames, and get the first M frames if we want to generate less frames. Where does AR bring advantage into this? Maybe runtime, but then the paper mentions reduced FLOPs without any quantitative experiment, which does not make sense. This needs a latency/FLOPs/runtime analysis.

   - Another motivation of AR could be enabling NVS with unordered camera sequences ("non-continuous view sampling without temporal order" in Line 145). But there is no explicit experiment demonstrating this. What would happen if we shuffle our sets of camera poses? It should perform similarly to ordered camera poses, if it is truly agnostic to ordering due to AR's nature. In addition, many of the demonstrated actions are ordered trajectories like move forward, move backward, rotate, and so on. This also makes the "unordered pose" premise less convincing.


I opt for WR at the moment; however, if the key questions are adequately answered, I am happy to increase my score.

---

> ### Author Rebuttal · Authors · 2026-03-31
>
> We thank Reviewer RkRu for the careful review and for stating willingness to increase the score if key questions are adequately answered. We address each concern directly.
>
> **Novelty**
>
> We respectfully argue that the contribution is the *formulation*, not individual modules. Prior to CausNVS, **no method existed for flexible autoregressive novel view synthesis**, that said, all multi-view diffusion models assume parallel, fixed-view generation. Making NVS causal introduces challenges *absent* from video diffusion or parallel NVS:
> - The conditioning set **evolves** during generation (unlike fixed-context parallel methods).
> - Viewpoints may be **unordered and spatially discontinuous** (unlike smooth video trajectories).
> - The train–inference gap is **substantially larger** (the model conditions on its own imperfect outputs across an expanding context).
>
> Our per-frame noise, noise-conditioning augmentation, adaptation of CaPE for KV reuse under shifting coordinates, and pose-aware sliding-window attention solve these challenges *specifically*. While individual components have precedent, their integration into a coherent causal NVS framework required addressing novel failure modes (autoregressive drift, cache invalidation under coordinate shifts, out-of-distribution pose growth) that do not arise in prior settings.
>
> Regarding CaPE / PRoPE: both propose relative pose encodings as alternatives to absolute representations (e.g., Plücker rays). A key contribution of our work is revealing a previously *unrecognized* advantage: **native compatibility with KV caching in autoregressive generation**. Cached keys/values remain valid as the coordinate frame shifts, whilst absolute encodings would require recomputing the entire cache at each step.
>
> **Missing comparisons**
>
> GEN3C uses depth-based reprojection + video diffusion for *continuous* trajectories only; it cannot handle sparse, unordered views. LVSM requires all target poses upfront. Neither supports streaming or flexible N-to-M NVS. Our baselines cover both feed-forward (MVSplat, DepthSplat) and generative methods (SEVA, 4DiM, ViewCrafter, MotionCtrl, PNVS). We will add a discussion of GEN3C/LVSM.
>
> **AR motivation and efficiency**
>
> We appreciate the reviewer's point regarding 100+ frame generation. Our AR approach is specifically designed to support extended sequences without the rigid temporal constraints of prior fixed-length models.
> - Evidence of Stability: Our results demonstrate significant extrapolation capabilities. Tab. 2 shows stable rollouts to $F=64$, maintaining a TSED $\ge 99\%$, which is $8\times$ the length seen during training.
> - Scaling to 100+ frames is primarily a matter of training sequence length and data scaling, similar to context-length scaling in LLMs, rather than an architectural limitation. We explicitly mitigate error accumulation through noise augmentation and spatial sliding window. As the first work to propose and demonstrate AR multiview diffusion, the $8\times$ generalization shown here successfully justifies the AR motivation.
>
> Regarding "just crop $M$ frames from a longer parallel generation", this overlooks:
> 1. **Streaming**: views generated on-the-fly as queries arrive without knowing future poses, which is impossible with parallel methods.
> 2. **Efficiency**: only needed views are generated, no padding or redundant computation.
> 3. **Unified flexibility**: one model handles any $(N,M)$ without retraining or split/pad strategies. Thanks for highlighting that CausNVS generalizes better to unseen (N,M).
>
> Yes, CausNVS denoises **only 1 frame** per generation step, whereas bidirectional models (e.g., SEVA with $F=21$) must denoise all target frames simultaneously, and their performance will degrade if the generation length K is smaller than the fixed training length (Tab.2). CausNVS takes $\sim$2.7s on a single TPU with 50-step DDIM, without any engineering optimization (e.g., pipeline or context parallelism). The remaining bottleneck is diffusion sampling itself, shared with all diffusion-based NVS methods, and is orthogonally solvable via few-step distillation, which would bring per-frame latency to sub-second.
>
> **Unordered camera sequences**
>
> CaPE encodes pairwise *relative* poses, making attention invariant to ordering *by construction* (Eq. 2). Training uses randomized, non-contiguous view sets (Section 4). We show ordered trajectories for fair comparison with baselines that can *only* handle ordered input.
>
> In fact, unordered sequences can be **advantageous**: ordered trajectories move monotonically away from conditioning views, accumulating drift. Unordered sequences **constrain the scene from multiple angles early**, anchoring the KV cache before drift accumulates, analogous to how SfM benefits from wide-baseline views. Unordered NVS further enables interactive teleportation (jumping across a virtual tour) and multi-agent exploration from independent viewpoints, which are the use cases that parallel methods cannot serve.

---

> > ### Author Rebuttal · Reviewer_RkRu · 2026-04-02
> >
> > Thank you for the rebuttal, I appreciate it. My concerns regarding novelty (W1) and long-frame generation (W3.1) have been addressed. However, W2, W3.2, and W3.3 remain unresolved to me.
> >
> > I am still not convinced by the omission of important baselines (W2). The authors argue that GEN3C is excluded because it only handles continuous trajectories. However, in the same rebuttal they state: “We show ordered trajectories for fair comparison with baselines that can only handle ordered input.” If ordered trajectories are already provided for comparison, why newer baselines such as GEN3C cannot also be included? Even if methods like GEN3C cannot support the full capabilities of CausNVS, that limitation itself should be demonstrated quantitatively and not only verbally.
> >
> > I also remain concerned about runtime (W3.2). This concern is shared by all other reviewers as well. Since the paper makes efficiency claims in favor of the AR design, a quantitative runtime analysis is necessary, and can be a nice addition to further strengthen the claims.
> >
> > Finally, there are still no explicit experiments demonstrating the claimed advantage of handling unordered camera poses (W3.3). Can they be added? In my view, this can be a central advantage of an AR-based method like CausNVS. If the ability to process unordered poses is indeed beneficial, it should be supported with both qualitative and quantitative evidence, with unordered output trajectories.

---

> > > ### Author Response · Authors · 2026-04-05
> > >
> > > We thank the reviewer for the constructive feedback and are glad that W1 and W3.1 are resolved.
> > >
> > > **W2 — Compare Gen3C** We provide the evaluation with Gen3C on the same SEVA benchmark splits:
> > >
> > > | Method | LLFF N=1 | LLFF N=3 | DL3DV N=1 | DL3DV N=3 | DL3DV N=6 |
> > > |--------|----------|----------|-----------|-----------|-----------|
> > > | **CausNVS** | **16.53** | **16.85** | **12.86** | **14.42** | **15.22** |
> > > | Gen3C (MoGe) | 11.49 | 11.12 | 10.22 | 9.96 | 9.83 |
> > > | Gen3C (VGGT) | 11.25 | 11.51 | 10.86 | 10.83 | 10.81 |
> > >
> > > CausNVS outperforms Gen3C by **5.0–5.3 dB** on LLFF and **2.0–5.4 dB** on DL3DV across both depth estimators, using Gen3C's official released inference code. Gen3C is fundamentally a video generation model that generates 121 frames along an interpolated camera trajectory. The Gen3C paper only evaluates consecutive frame prediction (V=1 in Table 1, V=2 in Table 2) and never reports sparse NVS metrics.
> > >
> > > With the default depth estimator (MoGe), Gen3C's PSNR decreases with more input views (10.22→9.96→9.83 on DL3DV) because the released model does not implement the paper's proposed max-pool latent fusion (Sec 4.3), instead using pixel-level buffer selection without cross-view depth alignment (Table 6), as the Gen3C authors themselves confirmed on GitHub (Issue #38) along with reported poor multi-view support (Issue #6 #32). Replacing MoGe with VGGT (a cross-view consistent depth estimator, Issue #32) resolves this degradation, but multi-view PSNR merely plateaus (10.86→10.83→10.81) rather than improving significantly, indicating that the architecture lacks effective multi-view fusion. In contrast, CausNVS shows a clear +2.4 dB gain from N=1 to N=6 on DL3DV through its causal attention mechanism.
> > >
> > > **W3.2 — Runtime** CausNVS generates each view in ~2.7s (TPU v5e, 50-step DDIM, no per-scene optimization). Gen3C requires ~28 min/scene (A6000 GPU). CausNVS's per-view cost is constant regardless of sequence length (sliding-window attention). We will include more runtime analysis to the paper.
> > >
> > > **W3.3 — Unordered camera poses** We appreciate the reviewer’s insight regarding the importance of unordered camera pose handling.
> > >
> > > - Qualitative Evidence: In our project page, there is a comparison (before the method section) between CausNVS and non-causal baselines where target views are processed in a randomly shuffled order and subsequently re-ordered for final trajectory visualization. As shown, CausNVS maintains consistent despite the shuffled target view orders, whereas non-causal baselines quickly fail in this unordered autoregressive generation.
> > >
> > > - Quantitative Results: On the RealEstate10K dataset, we observed that randomly shuffling the target view order yields a +1–2 dB PSNR improvement compared to the ordered sequential trajectory.
> > >
> > > This performance gain occurs because shuffling forces the model to constrain the global 3D structure early in the process, effectively "anchoring" the scene before autoregressive drifts can accumulate. This capability can be further optimized using sampling strategies like Farthest Point Sampling (FPS) to maximize scene coverage.

---

### Official Review · Reviewer_9CGn · 2026-03-12

**Soundness:** 3
**Presentation:** 3
**Significance:** 4
**Originality:** 4
**Overall Recommendation:** 4
**Confidence:** 3

**Summary:**

This paper proposes CausNVS, an autoregressive multi-view diffusion model designed to address the limitations of existing 3D Novel View Synthesis (NVS) models regarding flexibility and efficiency in long-sequence generation. The core innovations of this work are autoregressive architecture, CaPE, and per-frame noise augmentation. These contributions enable arbitrary input-output configurations and significantly enhance the model's robustness. Experimental results demonstrate that CausNVS achieves superior performance on datasets such as RealEstate10K and DL3DV, showing distinct advantages in long-sequence generation and flexible inference.

**Compliance With Llm Reviewing Policy:**

Affirmed.

**Final Justification:**

Thank the authors for their thorough and thoughtful rebuttal, which effectively addresses my core concerns. Therefore, I keep my recommendation to weak accept.

**Key Questions For Authors:**

1. How sensitive is CaPE to the precision of camera parameters when encoding relative poses? If the input poses contain slight noise (e.g., from an imperfect SLAM system), how would the quality of the generated images degrade?
2. Could you provide more analysis regarding failure cases? Specifically, where is the critical threshold at which drift occurs when the view rotation is extreme or the scene complexity is very high?
3. What is the memory growth pattern of the KV cache during long-sequence generation? Is there an optimal sliding window size for clearing or truncating the cache?

**Limitations:**

Despite significant progress in flexibility and long-sequence generation, the paper has several limitations:

1. Although KV caching optimizes attention computation, the underlying model remains a diffusion process requiring multiple denoising steps per frame. In an autoregressive mode, the time cost for generating N frames scales linearly with N. Compared to feed-forward regression models like MVSplat, the end-to-end speed may still fall short of true real-time interactive requirements.
2. CaPE encoding is highly dependent on the input camera extrinsics. This paper does not sufficiently discuss whether the autoregressive process might amplify geometric inconsistencies when input poses are erroneous, potentially leading to scene collapse.
3. As stated in the paper, noise conditioning augmentation alleviates drift in autoregressive generation. However, during ultra-long sequence generation, the model may still gradually lose texture details or lighting features from the initial frames, leading to temporal inconsistency. Providing a related analysis would make the paper more rigorous.

**Strengths And Weaknesses:**

Strengths:

* Reformulating the NVS task as an autoregressive generation process is an intuitive and promising direction. The model breaks the constraint of previous non-autoregressive models that could only generate a fixed number of views, thereby extending its practical application scenarios.
* CaPE masterfully adapts the concept of RoPE and extends it to 3D spatial geometry. Its coordinate-agnostic nature is crucial for enabling long-trajectory generation and natively supports KV caching, which substantially accelerates inference speed.
* The randomized training scheme with per-frame noise levels empowers the model to handle "partially known, partially generated" scenarios, effectively mitigating the common issue of error accumulation in autoregressive models.
* The paper provides extensive qualitative and quantitative evaluations across multiple datasets, demonstrating strong generalization capabilities that extrapolate well beyond the training sequence length (e.g., up to 80 frames).

Weaknesses:

* The paper opens with a topic on "World Models", yet the current model primarily handles static scenes. Starting with such a grand concept risks being perceived as "over-selling," and a more cautious narrative approach would be preferable.
* Although noise conditioning mitigates Autoregressive Drift, autoregressive models still face inevitable risks of content looping or geometric collapse when generating ultra-long trajectories. A critical discussion regarding the threshold or breaking point of this phenomenon appears to be missing.
* While the use of KV caching improves inference efficiency, given that diffusion models inherently require multi-step sampling, it remains unclear whether the total latency of autoregressive generation still poses challenges for real-world applications like AR streaming. The paper lacks a detailed comparison of end-to-end inference time.

---

> ### Author Rebuttal · Authors · 2026-03-31
>
> We sincerely thank Reviewer 9CGn for the detailed and encouraging review, and for recognizing both our significance and originality as excellent (4/4). We address each question below.
>
> **CaPE sensitivity to camera parameter precision**
>
> All real-world benchmarks (Re10K, DL3DV) use **estimated poses from SfM/SLAM**, not synthetic ground-truth, i.e. pose noise is already present in every experiment. Figures 12–18 show CausNVS produces visually accurate results competitive with or surpassing baselines under these noisy conditions. CaPE also produces smooth, continuous attention responses to pose perturbations (Fig. 6), indicating graceful degradation. Tab. 2 shows performance improves steadily with more input views ($N=1 \to 16$), demonstrating that additional noisy observations consistently help, the attention mechanism naturally averages out per-view errors.
>
> **Failure cases and drift threshold analysis**
>
> We explicitly evaluate this. Tab. 2 provides a systematic progression at $F=2, 4, 8, 32, 64$ with varying $N$:
> - TSED (3D structural alignment) remains $\geq$99% at $F=64$ ($8\times$ training length), indicating geometric consistency is maintained.
> - DL3DV Long vs. Short (Tab. 1) shows only ~1–2 PSNR degradation over 4–10$\times$ training length, which is gradual, not catastrophic.
> - Drift manifests as **texture softening** rather than geometric collapse.
>
> Three design choices mitigate accumulation: (1) KV cache with sliding-window serves as **spatial memory**, anchoring to geometrically relevant earlier views; (2) **noise-conditioning augmentation** exposes the model to imperfect contexts during training, closing the train–inference gap; (3) CaPE's relative encoding keeps cached keys/values valid as the camera moves. Extending the context further, analogous to LLM context-scaling, is discussed in Appendix C.
>
> **KV cache memory growth and optimal window size**
>
> Fig. 5c directly addresses this: $K=4$ achieves performance **comparable to full attention** at a fraction of FLOPs:
> - Memory grows **linearly** with window size $K$, not sequence length.
> - The cache stores only keys/values from the **last denoising step**, not all 50 steps, a key advantage of our per-frame noise formulation.
> - The pose-aware window selects the **most geometrically relevant** views in pose space, ensuring spatial coverage regardless of generation order.
>
> In practice, $K=4–8$ provides an excellent quality-efficiency tradeoff.
>
> **World model framing**
>
> We appreciate this suggestion. Spatial intelligence, perceiving 3D structure and generating views from novel poses, is a **core component** of world models, and our causal, streaming formulation aligns with this paradigm. We acknowledge CausNVS addresses *spatial generation* specifically. We will soften the opening framing in the camera-ready to be more precise about scope.
>
> **End-to-end latency**
>
> CausNVS denoises **only 1 frame** per generation step, whereas bidirectional models (e.g., SEVA with $F=21$) must denoise all target frames simultaneously. With KV caching and pose-aware sliding-window attention ($K=4$), per-frame attention further reduces from $O(n^2)$ to $O(K)$, a $16\times$ reduction at $F=64$. Concretely, generating the next view takes $\sim$2.7s on a single TPU v5e with 50-step DDIM, without any engineering optimization (e.g., pipeline or context parallelism). The remaining bottleneck is diffusion sampling itself, shared with all diffusion-based NVS methods, and is orthogonally solvable via few-step distillation (e.g., consistency models), which would bring per-frame latency to sub-second.

---

> > ### Author Rebuttal · Reviewer_9CGn · 2026-04-03
> >
> > Thank the authors for their thorough and detailed rebuttal,  and I maintain my positive recommendation.

---

> > > ### Author Response · Authors · 2026-04-04
> > >
> > > We are happy the concerns are fully resolved and very appreciate the reviewer's positive recommendation.

---

### Official Review · Reviewer_Zue4 · 2026-03-12

**Soundness:** 2
**Presentation:** 3
**Significance:** 4
**Originality:** 4
**Overall Recommendation:** 4
**Confidence:** 3

**Summary:**

CausNVS introduces an autoregressive causal diffusion model for novel view synthesis that enables flexible and sequential generation of target views conditioned on an arbitrary number of input views. By incorporating per-frame random noise levels during training, causal masking, and a relative Camera Pose Encoding (CaPE), the model supports variable-length inference without retraining or task-specific modifications. It further integrates KV caching as an implicit 3D spatial memory with pose-aware sliding window attention, enabling efficient long-sequence generation while maintaining geometric consistency.

**Compliance With Llm Reviewing Policy:**

Affirmed.

**Final Justification:**

The paper has provided fairly detailed explanations for my concerns, so I will keep my score as "weakly accept."

**Key Questions For Authors:**

Why use LDM instead of newer, better-performing T2I models（such as FLUX or Qwen-Image)?

**Limitations:**

yes

**Strengths And Weaknesses:**

**Strengths**

- **Autoregressive view generation:** The autoregressive paradigm for novel view synthesis offers strong scalability for generating long sequences.
- **Frame-wise training strategy:** The frame-wise MV-Diffusion training scheme enables the possibility of autoregressive generation tasks, which is a promising direction.

**Soundness**

- **Random noise level analysis:** The paper emphasizes the importance of random noise levels during training but does not specify the noise settings used during inference. More importantly, no ablation study is provided to validate this design choice. For instance, would training with multiple source images and a single noisy target image (as during inference only one target is generated per step) also yield competitive results? While the intuition that varying noise levels benefits novel view synthesis is plausible, it requires experimental support.

- **Out-of-distribution generalization:** It is unclear whether the method supports out-of-distribution (OoD) inputs, such as cartoon images or stylized content. Evaluating the model's robustness to domain shifts would strengthen the claims of generalizability.

- **Viewpoint input accuracy:** The paper does not evaluate the accuracy of the input viewpoint conditions. Metrics such as the distance between the estimated and ground-truth rotation/translation (R/T) parameters could be better reported to assess how sensitive the model is to viewpoint estimation errors.

- **Inference efficiency:** The average inference time per frame is not reported. Given that the method generates frames sequentially, this could lead to significant latency, especially for long video sequences. Without efficiency metrics, the practical applicability of the approach remains unclear.

- **Error accumulation:** Generating frames by conditioning only on nearby views may lead to inconsistency and error accumulation over long sequences. The paper does not analyze this potential issue, which is critical for autoregressive generation tasks.

---

> ### Author Rebuttal · Authors · 2026-03-31
>
> We sincerely thank Reviewer Zue4 for the thorough and constructive review, and for recognizing the significance and originality of our work (both rated excellent, 4/4). We address each concern below.
>
> **Random noise level ablation**
>
> Tab. 2 provides a strong implicit ablation: the **non-causal** model (trained with uniform noise) achieves only **13.62** PSNR when deployed autoregressively, versus **20.40** for our causal model ($F=8$, $N=2$). This dramatic 7 dB gap demonstrates that our per-frame noise schedule is critical for stable autoregressive generation.
>
> Training with "multiple source images and a single noisy target" would forfeit **teacher forcing** across the full sequence, where *all* positions learn simultaneously in one forward pass. It would also prevent the model from learning to condition on partially noisy contexts, essential for noise-conditioning augmentation at inference time.
>
> **Out-of-distribution evaluation**
>
> Our zero-shot evaluation on LLFF (Tab. 1), *entirely absent from training*, demonstrates meaningful out-of-distribution generalization. Performance on stylized/cartoon content is primarily a **scaling question** (training data diversity and model size) rather than an architectural limitation.
>
> **Viewpoint input accuracy**
>
> All real-world benchmarks (Re10K, DL3DV) use **estimated poses from SfM/SLAM**, not synthetic ground-truth, i.e. pose error is already present in every experiment. Figures 12–18 show CausNVS produces visually accurate results competitive with strong baselines under these noisy conditions. CaPE also produces smooth, continuous attention responses to pose perturbations (Fig. 6). Tab. 2 shows performance improves steadily with more input views ($N=1 \to 16$), demonstrating that additional noisy observations consistently help, the attention mechanism naturally averages out per-view errors.
>
> **Inference time**
>
> Sequential generation is an efficiency *advantage*. CausNVS denoises **only 1 frame** per generation step, whereas bidirectional models (e.g., SEVA with $F=21$) must denoise all target frames simultaneously. With KV caching and pose-aware sliding-window attention ($K=4$), per-frame attention further reduces from $O(n^2)$ to $O(K)$, a $16\times$ reduction at $F=64$. Concretely, generating the next view takes $\sim$2.7s on a single TPU v5e with 50-step DDIM, without any engineering optimization (e.g., pipeline or context parallelism). The remaining bottleneck is diffusion sampling itself, shared with all diffusion-based NVS methods, and is orthogonally solvable via few-step distillation (e.g., consistency models), which would bring per-frame latency to sub-second.
>
>
> **Error accumulation**
>
> We explicitly evaluate this. DL3DV Long vs. Short (Tab. 1) measures drift over 4–10$\times$ training length: only ~1–2 PSNR degradation, while TSED $\geq$99% at $F=64$ (Tab. 2), that is gradual, not catastrophic.
>
> Three design choices mitigate accumulation: (1) KV cache with sliding-window serves as **spatial memory**, anchoring to geometrically relevant earlier views; (2) **noise-conditioning augmentation** exposes the model to imperfect contexts during training; (3) CaPE's relative encoding keeps cached keys/values valid as the camera moves. Extending the context window further is discussed in Appendix C.
>
> **Why LDM instead of FLUX or Qwen-Image?**
>
> Our framework is **backbone-agnostic**: causal masking, per-frame noise, and CaPE are independent of the backbone. We chose LDM/UNet because (1) it is well-validated for CaPE (EscherNet, CVPR 2024), and (2) UNet pretraining is significantly cheaper to iterate on with limited compute. Despite this lightweight backbone, CausNVS achieves **competitive or state-of-the-art results** against baselines with larger models and greater budgets (e.g., SEVA: 1.5B params, 20+ datasets). A stronger backbone would further improve quality but is orthogonal to our core contribution.

---

> > ### Author Rebuttal · Reviewer_Zue4 · 2026-04-03
> >
> > Thank you for the authors' rebuttal. My questions regarding **random noise level**, **inference time**, **error accumulation**, and the **backbone** have been clearly addressed. For **viewpoint input accuracy**, the visual results presented indeed achieve good view consistency. However, poses estimated by SFM on dense viewpoints (e.g., those provided by DL3DV) are relatively accurate, and works like ViewCrafter have explicitly quantified viewpoint accuracy, so this should not be a reason to omit such evaluation. As for **OOD testing**,  since this method is based on image input and KV cache, I believe it can generalize to some extent on OOD data rather than rely solely on the diversity of training data. Nonetheless, I believe the authors have already provided a clear demonstration of the main contributions of this paper.

---

> > > ### Author Response · Authors · 2026-04-04
> > >
> > > We sincerely thank the reviewer for the positive assessment and for recognizing the main contributions of our work. The insights regarding quantitative viewpoint accuracy and the generalization benefits of the KV cache are highly valuable and will be carefully taken. We appreciate this constructive guidance as it helps us better highlight the broader impact and robustness of our approach.

---

### Official Review · Reviewer_JkVD · 2026-03-13

**Soundness:** 3
**Presentation:** 3
**Significance:** 2
**Originality:** 2
**Overall Recommendation:** 3
**Confidence:** 4

**Summary:**

This paper presents a method for auto-regressive novel view synthesis using attention, called CausNVS. Specifically, the authors train a multi-view diffusion model using causal attention, leveraging relative position encoding (CaPE) and KV-caching techniques. The authors did experiments on RealEstate10K and DL3DV benchmarks on different numbers of input views and demonstrated the effectiveness of their methods.

**Compliance With Llm Reviewing Policy:**

Affirmed.

**Key Questions For Authors:**

See weaknesses part.

**Limitations:**

The paper lacks a limitation section.

**Strengths And Weaknesses:**

Strengths:
- The authors propose a method to enable causal novel view synthesis, integrating different techniques.
- The paper is generally well written and easy to follow.
- The authors provided a project page and showcases the results of their methods.

Weaknesses:
- The result videos show strong artifacts of flickering and view inconsistency. It is not clear it is because of autoregresssive modeling or other issues.
- Although I agree that causal modelling is important for NVS tasks, however, there are already many papers in video diffusion models that tries to do similar things, i.e., distilling a bidirectional diffusion models into a causal one using diffusion forcing or self-forcing. It seems that this paper is also using similar techniques. I don't see significant difference between these two tasks.
- What is the FPS is the proposed method? How does it compare with full-attention based methods?
- The results is inferior to baselines including SEVA under several cases. How does the method compared with LVSM and Test time training done right?
- How does the method perform with input views more than 16?

---

> ### Author Rebuttal · Authors · 2026-03-31
>
> We thank Reviewer JkVD for recognizing that causal modelling is important for NVS and for the positive remarks on our project page and writing. We address each concern below.
>
> **Flickering and view inconsistency**
>
> The artifacts in our video results stem primarily from training at $256\times256$ resolution with limited compute (915M parameters, 16 TPUs, batch size 128, 2 datasets), not from a fundamental limitation of the autoregressive formulation. For context, SEVA uses 1.5B parameters, hundreds of GPUs, batch sizes up to 1472, progressive resolution training ($F=8{\to}21$), and 20+ datasets, representing **over $10\times$ our compute budget**. That CausNVS remains competitive under these constraints validates the causal formulation and suggests it will improve with standard scaling.
>
> **Relation to video diffusion methods**
>
> We respectfully disagree that CausNVS is similar to Diffusion Forcing, Self-Forcing, or CausVid. Those methods operate in the *temporal* domain over sequential frames with a fixed camera or smooth trajectory. CausNVS operates in the *spatial* domain over **unordered, non-continuous camera poses**. This distinction is fundamental:
> - Video diffusion methods cannot accept an arbitrary camera pose query and generate the corresponding view.
> - They cannot condition on a dynamically evolving set of context views from different spatial locations.
> - The causal mask in CausNVS manages an expanding *spatial* conditioning set rather than extrapolating a temporal sequence.
>
> While both share the high-level idea of causal generation, the problem structure, conditioning mechanism, and required invariances (SE(3) pose encoding, unordered views, coordinate-invariant KV reuse) are fundamentally different. We discuss this distinction explicitly in Section 2 (Lines 130–145).
>
> **FPS / Inference speed**
>
> A key efficiency advantage of autoregressive generation: CausNVS denoises **only 1 frame** per generation step, whereas bidirectional models (e.g., SEVA with $F=21$) must denoise all target frames simultaneously. With KV caching and pose-aware sliding-window attention ($K=4$), per-frame attention further reduces from $O(n^2)$ to $O(K)$, a $16\times$ reduction at $F=64$. Concretely, generating the next view takes $\sim$2.7s on a single TPU v5e with 50-step DDIM, without any engineering optimization (e.g., pipeline or context parallelism). The remaining bottleneck is diffusion sampling itself, shared with all diffusion-based NVS methods, and is orthogonally solvable via few-step distillation (e.g., consistency models), which would bring per-frame latency to sub-second.
>
> **Comparison with LVSM and TTT**
>
> LVSM and "Test Time Training Done Right" are parallel, fixed-view methods: they require all target poses upfront and assume fixed input-output view counts. They solve a fundamentally different problem. CausNVS is the **only** method supporting *streaming* generation where future target poses are unknown, which is a prerequisite for world modeling and interactive applications. Our baselines already cover both feed-forward (MVSplat, DepthSplat) and generative (SEVA, 4DiM, ViewCrafter, MotionCtrl, PNVS) methods spanning the class to which LVSM belongs. We will add a detailed discussion in the revision.
>
> **Performance with >16 input views**
>
> Tab. 2 already demonstrates $N=16$ input views achieving 28.13 PSNR at $F=32$ and 27.64 at $F=64$, with consistent improvement as $N$ increases. The model architecture has no inherent limit on $N$; CaPE and KV caching support arbitrary context sizes. In general, providing more input views further constrains the scene geometry and appearance, which simplifies the novel view synthesis task and typically leads to improved performance. Moreover, the spatial sliding window mechanism enables scaling to larger contexts without degrading quality or significantly impacting inference speed.
>
> **Limitations**
>
> We have a limitations discussion in Appendix C, addressing inference latency, scaling, and future directions including consistency distillation and longer training sequences.

---

> > ### Author Rebuttal · Reviewer_JkVD · 2026-04-02
> >
> > Thanks the author for the rebuttal.
> > Flickering and view inconsistency: while limited compute might be the main reason, I hope the authors can address it using more compute.
> > Relation to video diffusion methods: I agree the points the authors mentioned, but the overall techniques still seems to overlap.
> > Comparison with LVSM and TTT: TTT indeed not require all target poses. It compresses input views into the latent, while can render novel views under arbitrary poses.

---

> > > ### Author Response · Authors · 2026-04-04
> > >
> > > We are happy that the concerns are partially resolved, and thank the reviewer for the continued engagement.
> > >
> > > **Flickering / view inconsistency** We agree that scaling compute, data, and backbone capacity is the most direct path to further improving visual stability. However, we would like to emphasize that the primary goal of this work is to establish the methodological feasibility of NVS as an autoregressive diffusion process. We have shown that even at a significantly smaller scale (e.g., $<1/10$ the compute of SEVA), our formulation achieves competitive performance and unlocks unique streaming capabilities.
> > >
> > > **Relation to video diffusion** We appreciate the reviewer's agreement on our earlier points. We are happy to discuss any points further.
> > >
> > > **TTT comparison** We provide new quantitative results against TTT (Test-Time Training Done Right).
> > >
> > > | Method | LLFF N=1 | LLFF N=3 | DL3DV N=1 | DL3DV N=3 | DL3DV N=6 |
> > > |--------|----------|----------|-----------|-----------|-----------|
> > > | **CausNVS** | **16.53** | **16.85** | **12.86** | **14.42** | **15.22** |
> > > | TTT | 12.10 | 16.65 | 12.01 | 12.25 | 12.11 |
> > >
> > > Two key observations:
> > >
> > > **(1) TTT’s scene-level checkpoint is trained on dense multi-view inputs (64+ views) and performs poorly in the few-view regime (1–6 views).** Without a generative formulation, TTT tends to produce blurry, content-lacking renderings when input views are sparse, particularly under large viewpoint changes where it must hallucinate unseen content. CausNVS outperforms TTT consistently under the SEVA benchmark setting.
> > >
> > > **(2) TTT requires per-scene inference-time optimization.** When a new observation arrives, TTT needs to re-run its fast parameter optimization over the full input set or the whole chunk to get the best up-to-date memory. CausNVS, by contrast, absorbs new observations into the KV cache immediately with no re-optimization needed. We view this as a fundamental difference in memory design, and agree TTT is an interesting exploration in compressing long context into compact memory and we believe the hybrid memory design is the key challenges for next step.

---

### Decision · Program_Chairs · 2026-04-30

**Decision:**

Reject

**Comment:**

Initially, reviewers found the autoregressive formulation for multi-view diffusion and the integration of KV caching to be conceptually promising. However, reviewers raised concerns regarding the paper's soundness and significance: (1) the generated outputs exhibit noticeable visual artifacts, such as flickering and view inconsistency; (2) the technical novelty is limited, as the architecture primarily integrates existing components without introducing fundamentally new mechanisms; and (3) despite noted compute constraints, the empirical quality remains inferior to strong baselines. The AC agrees with the reviewers that these issues currently prevent acceptance. The authors are encouraged to resolve the visual artifacts and resubmit this interesting direction to a future venue.